# Postsynaptic plasticity of cholinergic synapses underlies the induction and expression of appetitive and familiarity memories in *Drosophila*

**Carlotta Pribbenow[1], Yi-chun Chen[1†], M-Marcel Heim[1†], Desiree Laber[1†], Silas Reubold[1†], Eric Reynolds[1†], Isabella Balles[1], Tania Fernández-d V Alquicira[1], Raquel Suárez-Grimalt[1,2], Lisa Scheunemann[1,3,4], Carolin Rauch[1], Tanja Matkovic[4], Jörg Rösner[5], Gregor Lichtner[1,6], Sridhar R Jagannathan[1], David Owald[1,2,3]***

[1]Institute of Neurophysiology, Charité – Universitätsmedizin Berlin, corporate member of Freie Universität Berlin and Humboldt-Universität zu Berlin, and Berlin Institute of Health, Berlin, Germany; [2]Einstein Center for Neurosciences Berlin, Berlin, Germany; [3]NeuroCure, Charité – Universitätsmedizin Berlin, corporate member of Freie Universität Berlin and Humboldt-Universität zu Berlin, and Berlin Institute of Health, Berlin, Germany; [4]Institut für Biologie, Freie Universität Berlin, Berlin, Germany; [5]NWFZ, Charité – Universitätsmedizin Berlin, corporate member of Freie Universität Berlin and Humboldt-Universität zu Berlin, and Berlin Institute of Health, Greifswald, Germany; [6]Universitätsmedizin Greifswald, Department of Anesthesia, Critical Care, Emergency and Pain Medicine, Greifswald, Germany

**\*For correspondence:**
david.owald@charite.de

†These authors contributed equally to this work

**Competing interest:** The authors declare that no competing interests exist.

**Abstract** In vertebrates, several forms of memory-relevant synaptic plasticity involve postsynaptic rearrangements of glutamate receptors. In contrast, previous work indicates that *Drosophila* and other invertebrates store memories using presynaptic plasticity of cholinergic synapses. Here, we provide evidence for postsynaptic plasticity at cholinergic output synapses from the *Drosophila* mushroom bodies (MBs). We find that the nicotinic acetylcholine receptor (nAChR) subunit α5 is required within specific MB output neurons for appetitive memory induction but is dispensable for aversive memories. In addition, nAChR α2 subunits mediate memory expression and likely function downstream of α5 and the postsynaptic scaffold protein discs large (Dlg). We show that postsynaptic plasticity traces can be induced independently of the presynapse, and that in vivo dynamics of α2 nAChR subunits are changed both in the context of associative and non-associative (familiarity) memory formation, underlying different plasticity rules. Therefore, regardless of neurotransmitter identity, key principles of postsynaptic plasticity support memory storage across phyla.

## Editor's evaluation

Learning-dependent plasticity is thought to take place predominantly presynaptically in Drosophila. This paper by the Owald group adds important evidence for postsynaptic plasticity mechanisms, including that appetitive memory is impaired when nAChR subunits (α2, α5) and scaffold protein Dlg are downregulated in specific mushroom body output neurons. In a tour-de-force, they combine physiology, *Drosophila* genetics, and behaviour and the work also emphasises the similarities in learning and memory mechanisms between vertebrates and invertebrates.

## Introduction

The efficacy of synaptic transmission, also referred to as synaptic weight, can be increased or decreased following changes in neural activity profiles or concurrent action of neuromodulators, such as dopamine. Resulting changes to how synapses relay information underlie synaptic plasticity, which is widely believed to be the basis of memory storage (*Glanzman, 2010*; *Korte and Schmitz, 2016*; *Nicoll, 2017*). While, it is generally accepted that synaptic plasticity can serve as memory substrate from flies to humans, it is unclear to what degree neurophysiological and molecular principles underlying synaptic plasticity are evolutionarily conserved. One main difference between vertebrates and invertebrates is that memory-storing synapses in vertebrates use glutamate as their primary transmitter, while those in invertebrates (at least for *Drosophila melanogaster* and *Sepia officinalis*) use acetylcholine (*Barnstedt et al., 2016*; *Owald and Waddell, 2015*; *Shomrat et al., 2011*). Furthermore, it is widely established that invertebrate nervous systems utilize presynaptic plasticity (with plasticity referring to changes leading to either strengthening (potentiation) or weakening (depression) of synaptic transmission and the rearrangement or exchange of synaptic molecules underlying changed transmission) for storing memories, while the degree to which postsynaptic plasticity can be used is less clear. In contrast, it is well established (*Glanzman, 2010*; *Korte and Schmitz, 2016*; *Nicoll, 2017*) that storing information in vertebrates can depend on both pre- and postsynaptic mechanisms, including postsynaptic rearrangements of neurotransmitter receptors.

Detailed knowledge of the anatomical wiring and functional signaling logic of the *Drosophila* learning and memory centers, the mushroom bodies (MBs; third ['higher']-order brain center; learning takes place three synapses downstream of sensory neurons; *Owald and Waddell, 2015*; *Owald et al., 2015*; *Aso et al., 2019*; *Aso et al., 2014b*; *Aso et al., 2014a*; *Bouzaiane et al., 2015*; *Cohn et al., 2015*; *Felsenberg et al., 2018*; *Felsenberg et al., 2017*; *Hattori et al., 2017*; *Ichinose et al., 2015*; *Lewis et al., 2015*; *Pai et al., 2013*; *Perisse et al., 2016*; *Séjourné et al., 2011*; *Plaçais et al., 2013*) allow one to address to what extent synaptic mechanisms underlying memory storage are comparable across evolution, despite the use of different neurotransmitter systems. The weights of Kenyon cells (KCs) to MB output neuron (MBON) synapses are modulated by dopaminergic neurons (DANs), which anatomically divide the MBs into at least 15 functional compartments. At the level of these compartments, information is stored on appetitive and aversive (odor) associations, in addition to non-associative information, such as the relative familiarity of an odor (*Owald and Waddell, 2015*; *Owald et al., 2015*; *Aso et al., 2019*; *Aso et al., 2014b*; *Aso et al., 2014a*; *Bouzaiane et al., 2015*; *Cohn et al., 2015*; *Hattori et al., 2017*; *Hige et al., 2015*; *Takemura et al., 2017*), a distinct form of habituation. Summed up (*Owald and Waddell, 2015*), output from the individual MB compartments will give rise to specific behaviors, weighing up appetitive and aversive associations as well as, for instance, the familiarity of a stimulus.

Studies so far have identified several traits pointing toward presynaptic storage mechanisms within the KCs during memory formation (*Bilz et al., 2020*; *Boto et al., 2014*; *Handler et al., 2019*; *Ehmann et al., 2018*). Indeed, some studies that have blocked neurotransmitter release from KCs during learning *Dubnau et al., 2001*; *McGuire et al., 2001*; *Schwaerzel et al., 2002* have brought postsynaptic contributions to synaptic plasticity into question.

In vertebrates, typically, postsynaptic long-term changes (*Korte and Schmitz, 2016*; *Kandel et al., 2014*; *Lüscher and Malenka, 2011*) are mediated via NMDA-sensitive glutamate receptors (NMDAR) that induce ('induction'; *Nicoll, 2017*) an expression phase ('expression'; *Nicoll, 2017*) through changed glutamatergic AMPA receptor (AMPAR) dynamics in dependence of postsynaptic scaffolds like PSD-95 (*Won et al., 2017*). Invertebrate nAChRs in principle could take over similar functions to their glutamatergic counterparts in vertebrates, despite their differing molecular characteristics (*Thompson et al., 2010*). Indeed, nAChRs are pentamers that can be composed of homomeric assemblies of α subunits or heteromeric combinations of different α and β subunits. The composition of subunits determines the physiological properties of the nAChRs (*Thompson et al., 2010*; *Dent, 2010*; *Ihara et al., 2020*; *Lansdell et al., 2012*), and synaptic weights could, in theory, be adjusted through the exchange of receptor subunits or entire complexes.

Here, we capitalize on the genetic accessibility to individual output neurons of the MBs to directly test whether postsynaptic receptors play a role in memory storage. Derived from combined neurophysiological, behavioral, light microscopic, and molecular approaches, our data are supportive of a sequential role for nAChR subunits in appetitive memory storage at the level of MBONs. Using

artificial training protocols, we demonstrate that postsynaptic calcium transients can change in response to concurrent activation of DANs and application of acetylcholine, circumventing KC output. Blocking KC output during appetitive, but not aversive, learning abolishes memory performance. Moreover, specific knockdown of the α5 nAChR subunit, but none of the other six α subunits, in the M4/6 MBONs (also known as MBON-γ5β'2a, MBON-β'2mp, MBON- β2β'2a, and MBON-β'2mp bilateral) – an output junction involved in coding appetitive and aversive memories (*Owald et al., 2015*; *Bouzaiane et al., 2015*; *Lewis et al., 2015*) – impairs immediate appetitive memories. Knockdown of α2 or α5, however, interferes with 3-hr appetitive memories, as does knockdown of the scaffold discs large (Dlg). We report differential distribution of α subunits throughout the MB and demonstrate that signal recovery of GFP (green fluorescent protein)-tagged subunits (as measured through fluorescence signal recovered after photobleaching) is changed through plasticity protocols. In addition, postsynaptically expressed non-associative familiarity learning that is encoded at the level of the α'3 neurons of the MBs also depends on α5 and α2 signaling as well as α2 dynamics. We propose a temporal receptor model and speculate that, in *Drosophila*, nAChR subunits α5 and α2 take roles similar to NMDAR and AMPAR in vertebrates for memory induction and expression, indicating that the general principle for postsynaptic plasticity independent of the neurotransmitter system used could be conserved throughout evolution.

## Results

### Neurotransmitter release from KCs is required for appetitive learning

If the postsynapse need not see the neurotransmitter during training, it would likely be dispensable for memory induction. One key argument in favor of exclusively presynaptic memory storage mechanisms in *Drosophila* is based on experiments suggesting that blocking KC or KC subset output selectively during (olfactory) learning leads to unaltered or mildly changed memory performance (*Dubnau et al., 2001*; *Krashes et al., 2007*; *McGuire et al., 2001*; *Schwaerzel et al., 2002*). However, other studies have reported memory impairments following KC subset or downstream circuit element block during training (*Krashes et al., 2007*; *Yamazaki et al., 2018*; *Ichinose et al., 2021*). Moreover, protein synthesis was shown to be required at the level of MBONs for long-term memory formation (*Pai et al., 2013*; *Widmer et al., 2018*; *Wu et al., 2017*). To corroborate that the postsynapse or downstream circuits would need to 'see' the neurotransmitter for memory storage, we first revisited experiments blocking KC output selectively during T-maze training, exposing the animals either to sugar-odor or shock-odor pairings (*Figure 1a–b*, *Figure 1—figure supplement 1a-b*).

We expressed the temperature-sensitive *dynamin* mutant UAS-*Shibire*ᵗˢ (Shi) at the level of KCs (R13F02-Gal4), trained animals at the restrictive temperature (32°C), and tested for memory performance at permissive temperature (23°C) 30 min later. These manipulations allowed us to interfere with the synaptic vesicle exo-endocycle specifically during conditioning, while reinstating neurotransmission afterward: by choosing the 30-min time point, we made sure to restore functional Dynamin and not to interfere with any process underlying memory retrieval. Consistent with previous reports (*Dubnau et al., 2001*; *McGuire et al., 2001*; *Schwaerzel et al., 2002*), a slight drop in aversive memory performance (*Figure 1a*) was not statistically different from their genetic controls, and also observable in the permissive temperature controls (23°C; see *Figure 1—figure supplement 1a*). In contrast, memories were completely abolished following block of KC output during appetitive training (*Figure 1b*, *Figure 1—figure supplement 1b*).

While it remained unclear as to how far postsynaptic plasticity at the KC to MBON synapse could be involved in memory storage, several lines of evidence have implicated circuit mechanisms downstream of KCs to be involved in memory formation (*Pai et al., 2013*; *Ichinose et al., 2021*; *Widmer et al., 2018*; *Wu et al., 2017*). We next asked whether the requirement for neurotransmission during appetitive learning was specific to the KC output synapse. To do so, we took an analogous approach, this time blocking neurotransmission from downstream M4/6 (MBON-γ5β'2a, MBON-β'2mp, MBON-β2β'2a, and MBON-β'2mp bilateral) MBONs (MB011B Split-Gal4) during appetitive training. We focused on the M4/6 set of MBONs as blocking these during memory retrieval crucially interferes with appetitive memory expression, while, on a physiological level, memory-related plasticity is observable (*Owald and Waddell, 2015*; *Owald et al., 2015*; *Felsenberg et al., 2018*). When blocking M4/6 during appetitive training, but not retrieval, memory scores were similar to those of control groups

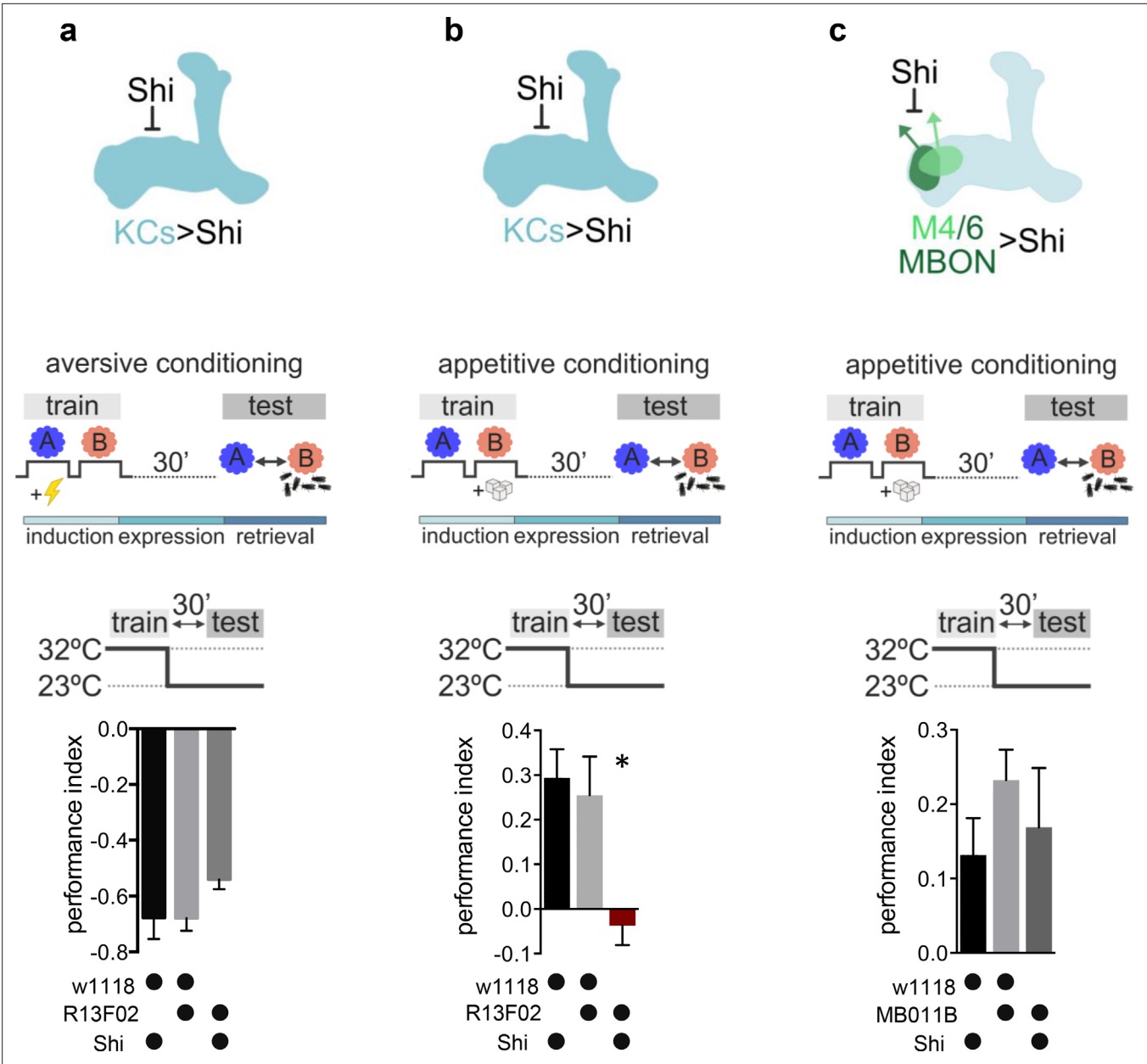

**Figure 1.** Kenyon cell (KC) neurotransmitter release is required for the acquisition of appetitive memories. (**a–c**) Flies expressing temperature-sensitive Shibire (Shi) within KCs or mushroom bodies output neurons (MBONs) are trained at restrictive temperature (32°C), and subsequently placed at permissive temperature (23°C) throughout the consolidation and retrieval phase. Memory performance was tested 30 min after training at permissive temperature. Shi blocks neurotransmitter release at 32°C. (**a**) Block of neurotransmitter release from KCs (driver line R13F02-Gal4) during training does not impact 30-min aversive memory performance. Bar graphs: mean ± SEM; n=7–8; one-way ANOVA followed by Tukey's test (p>0.05). (**b**) Block of neurotransmitter release from KCs (driver line R13F02-Gal4) during training impairs 30-min appetitive memory performance. Bar graphs: mean ± SEM; n=10–16; Kruskal-Wallis followed by Dunn's test (p<0.05), *p<0.05. (**c**) Block of neurotransmitter release from M4/6 MBONs (driver line MB011B [Split-GAL4]) during training does not impact 30-min appetitive memory performance. Bar graphs: mean ± SEM; n=14–24; one-way ANOVA followed by Tukey's test (p>0.05). Also see *Figure 1—figure supplement 1* for further information.

The online version of this article includes the following figure supplement(s) for figure 1:

**Figure supplement 1.** Permissive temperature controls accompanying *Figure 1*.

(*Figure 1c*), suggesting that the sites of plasticity are likely to be the KC to MBON synapse in general, with one major site specifically being the connections between KCs and M4/6 MBONs.

Thus, our experiments suggest that neurotransmitter release from KCs during training is required for the formation of appetitive memories but is less crucial for the formation of aversive memories.

## The α5 nAChR subunit is required for induction and α2 for expression of appetitive memories

Requirement for presynaptic neurotransmitter release alone does not necessarily mean that postsynaptic plasticity is involved in appetitive memory formation. To address a putative postsynaptic role in memory formation, we next interfered with the postsynaptic receptor composition. Given that KCs are cholinergic, we screened for memory requirement of all nicotinic α subunits at the level of the M4/6 MBONs (MB011B Split-Gal4; *Figure 2*) using previously published (*Barnstedt et al., 2016*; *Cervantes-Sandoval et al., 2017*) genetically targeted RNAi (UAS-nAChR^RNAi, please see Methods for detailed lines). We concentrated on the nAChR α subunits, as they are crucial components for all possible heteromeric or homomeric receptor pentamers (*Dent, 2010*).

When flies were tested for immediate appetitive memory, only knockdown of the α5 subunit produced performance that was statistically different from the controls (*Figure 2a*, *Figure 2—figure supplement 1a*, *Figure 2—figure supplement 2e*). Testing 3-hr appetitive memory performance revealed a significant memory impairment in flies with α1, α2, and α5 knockdown (*Figure 2b*, *Figure 2—figure supplement 1b*, *Figure 2—figure supplement 2f-h*). While α5 subunits can form homomeric channels (*Lansdell et al., 2012*), α1 and α2 can partake in heteromeric channels together (*Ihara et al., 2020*). We therefore concentrated on the α5 and α2 nAChR subunits in subsequent analyses.

To exclude developmental contributions to the observed memory defects, we repeated the immediate and 3-hr appetitive memory experiments for α5 as well as the 3-hr appetitive memory experiments for α2 knockdown animals, while suppressing RNAi expression (VT1211-Gal4>UAS-nAChR^RNAi) using the temperature-sensitive Gal4 repressor Gal80^ts (tubP-GAL80^ts) during development (<20°C), up until 3–5 days (32°C) before memory testing. Memory impairments were confirmed in all cases (*Figure 2e–g*; 32°C) but not detected in temperature controls where the RNAi expression was suppressed throughout (<20°C; *Figure 2—figure supplement 2a-c*).

We also tested aversive immediate and 3-hr memory using the same genetic settings (*Figure 2c and d*, *Figure 2—figure supplement 1c,d*). None of the knockdowns differed significantly from controls, with the exception of α7 at the 3-hr time point. As, comparable to vertebrate systems, α7 also plays a significant role at presynaptic neurites (*Eadaim et al., 2020*), we did not follow up on this observation in this study.

As M4/6 output is also required for appropriate aversive memory expression (*Owald et al., 2015*; *Bouzaiane et al., 2015*), α2 and α5 knockdown not impacting aversive memory performance suggested that the observed appetitive memory impairments were not simply a consequence of lost postsynaptic responsiveness to acetylcholine. To further corroborate this, we turned to a brain explant preparation and applied acetylcholine focally to the dendrites of M4/6 neurons (VT1211-Gal4) that expressed the calcium indicator UAS-GCaMP6f, for both control and knockdown settings, in the presence of the blocker of voltage-gated sodium channels TTX (*Barnstedt et al., 2016*; *Raccuglia et al., 2019*). Dendritic calcium transients were comparable between all groups (*Figure 3—figure supplement 1e*). We also observed presynaptic calcium transients in all genotypes (*not shown*) after applying acetylcholine to the presynaptic MBON boutons, making presynaptic deficits following α2 or α5 knockdown unlikely.

Therefore, we conclude that, at the level of M4/6 neurons, immediate and 3-hour appetitive memories are affected by knockdown of the α5 subunit, whereas 3-hr memories also require the presence of α1- and α2-bearing receptors in addition. The observed temporal profile of requirement for memory of α1- and α2-bearing receptors relative to those incorporating the α5 subunit could potentially point to a temporal sequence of receptor function during initial memory formation and subsequent memory expression.

## The postsynaptic scaffold Dlg is required for 3-hr appetitive memory

At mammalian glutamatergic synapses, postsynaptic receptor-mediated changes in synaptic weight (the efficacy of neurotransmitter-mediated signal propagation) rely on receptor stabilization or destabilization that can be mediated via scaffolding molecules. One such scaffold, PSD-95, that is mostly involved in AMPAR motility, is conserved at *Drosophila* synapses. The ortholog Dlg (*Bachmann et al., 2004*; *Soukup et al., 2013*) is expressed throughout the brain, with MB compartment-specific enrichment noted previously (*Crittenden et al., 1998*; also compare Figure 4a,b). We investigated

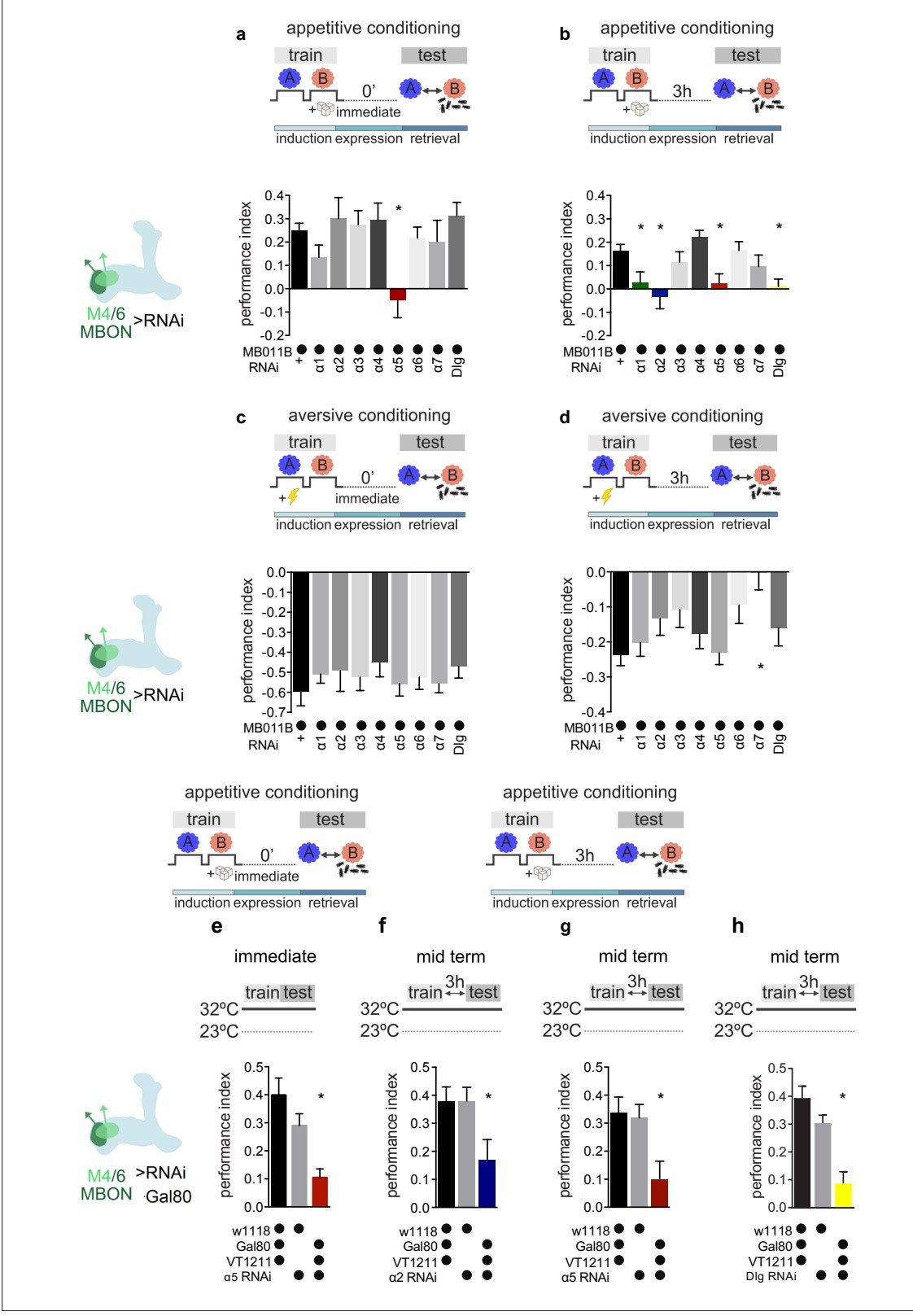

**Figure 2.** Specific nicotinic acetylcholine receptor (nAChR) α subunits are needed for specific memories in M4/6 neurons. Training and testing protocols indicated schematically. A and B indicate different odors. (**a**) Immediate appetitive memories are impaired following RNAi knockdown of the α5 nAChR subunit in M4/6 mushroom bodies output neurons (MBONs; driver line MB011B [Split-GAL4]). Bar graphs: mean ± SEM; n=8–13, for controls: n=20; one-way ANOVA followed by Dunnett's test (p<0.05), *p<0.05. Note: data depicted correspond to initial screen, please see *Figure 2—figure supplement*

*Figure 2 continued on next page*

*Figure 2 continued*

**2e** for alternate display including all genetic controls. (**b**) RNAi knockdown of the α1, α2, α5 nAChR subunits or discs large (Dlg) in M4/6 MBONs (driver line MB011B [Split-GAL4]) impairs 3-hr appetitive memories. Bar graphs: mean ± SEM; n=12–26, for controls: n=38; Kruskal-Wallis followed by Dunn's test (p<0.05), *p<0.05. Note: data depicted correspond to initial screen, please see *Figure 2—figure supplement 2* f–i for alternate display including all genetic controls. (**c**) Immediate aversive learning is not impaired by RNAi knockdown of any subunit in M4/6 MBONs (driver line MB011B [Split-GAL4]). Bar graphs: mean ± SEM; n=6–8, for controls: n=12; Kruskal-Wallis followed by Dunn's test (p>0.05). (**d**) 3-hr aversive memory is not affected by knockdown of α subunits with the exception of α7 (driver line MB011B [Split-GAL4]). Bar graphs: mean ± SEM; n=21–32, for controls: n=61. Kruskal-Wallis followed by Dunn's test (p<0.05), *p<0.05. (**e**) RNAi knockdown of the α5 subunit in M4/6 MBONs (driver line VT1211-Gal4) is suppressed during development using Gal80^ts. 3–5 days before the experiment RNAi knockdown was induced. Immediate memory is significantly impaired. Bar graphs: mean ± SEM; n=6–7; Kruskal-Wallis followed by Dunn's test (p<0.05), *p<0.05. (**f**) RNAi knockdown of the α2 subunit in M4/6 MBONs (driver line VT1211-Gal4) is suppressed during development using Gal80^ts. 3–5 days before the experiment RNAi knockdown was induced. 3-hr memories are significantly impaired. Bar graphs: mean ± SEM; n=16–17; one-way ANOVA followed by Tukey's test (p<0.05), *p<0.05. (**g**) RNAi knockdown of the α5 subunit in M4/6 MBONs (driver line VT1211-Gal4) is suppressed during development using Gal80^ts. 3–5 days before the experiment RNAi knockdown was induced. 3-hr memories are significantly impaired. Bar graphs: mean ± SEM; n=25–27; one-way ANOVA followed by Tukey's test (p<0.05), *p<0.05. (**h**) RNAi knockdown of Dlg in M4/6 MBONs (driver line VT1211-Gal4) is suppressed during development using Gal80^ts. 3–5 days before the experiment RNAi knockdown was induced. 3-hr memories are significantly impaired. Bar graphs: mean ± SEM; n=8–11; one-way ANOVA followed by Tukey's test (p<0.05), *p<0.05. Also see *Figure 2—figure supplements 1 and 2* for further information.

The online version of this article includes the following figure supplement(s) for figure 2:

**Figure supplement 1.** Genetic controls and alternate data display of data presented in *Figure 2*.

**Figure supplement 2.** Genetic controls and alternate data display of data presented in *Figure 2*.

appetitive and aversive memory performance following M4/6-specific knockdown of Dlg (*Figure 2*, *Figure 2—figure supplement 1a-d*; MB011B Split Gal4 >UAS-Dlg^RNAi). Performance scores comparable to controls were found for both immediate appetitive and aversive memories (*Figure 2a,c and d*, *Figure 2—figure supplement 1a,c and d*). Dlg knockdown (VT1211-Gal4>UAS-Dlg^RNAi), however, specifically abolished 3-hr appetitive memory performance (*Figure 2b*, *Figure 2—figure supplements 1b and 2i*), while Gal80^ts experiments excluded a developmental defect (*Figure 2h*, *Figure 2—figure supplement 2d*). The temporal profile of Dlg requirement therefore closely matched that of α2 nAChR subunits.

## Bypassing the presynapse: induction of persistent associative plasticity in the postsynaptic compartment

Recent ultrastructural data has revealed direct synaptic connections between DANs and MBONs (*Takemura et al., 2017*; *Eichler et al., 2017*), giving rise to a motif that could allow for postsynaptic plasticity induction (reflected by the lasting change of synaptic weights; see schematic in *Figure 3a*). In order to directly test whether postsynaptic plasticity could take place at the level of MBONs, we next conducted neurophysiological proof-of-principle experiments.

To minimize potential non-associative effects on synaptic properties induced by acute sensory experiences or general network activity, we used an explant brain preparation bathed in TTX from flies expressing the red light-activatable channelrhodopsin CsChrimson (lexAop-CsChrimson^tdTomato) in a subset of DANs (PAM neurons; R58E02-LexA) and the calcium indicator GCaMP6f (UAS-GCaMP6f) in M4/6 MBONs (MB011B Split Gal4).

While dopamine release was controlled by red light flashes, neurotransmitter release from KCs was mimicked by focal pressure ejection of acetylcholine to the dendrites of the M6 (MBON-γ5β'2a) MBON (M6 was chosen for technical reasons, as these neurons are most accessible for the used imaging technique). We verified that KC presynapses do not respond to acetylcholine application (*Barnstedt et al., 2016*), using both calcium imaging and imaging of synaptic vesicle exocytosis at the level of KC axons (*Figure 3c* and *Figure 3—figure supplement 1f,g*). The observed absence of KC activation, with acetylcholine being applied from an external source (*Figure 3a*), minimized noise attributable to possible presynaptic contributions.

Our protocols consisted of training phases where we differentiated between temporal pairing of acetylcholine and optogenetic activation of DANs ('paired', *Figure 3b and h*), dopamine only ('red-light only', *Figure 3b, e and g*), or 'acetylcholine only' (*Figure 3b, d and f*). Acetylcholine application preceded (pre) and followed each training step (post) to assess potential synaptic weight changes following training ('testing'). We found that test responses were significantly elevated following the

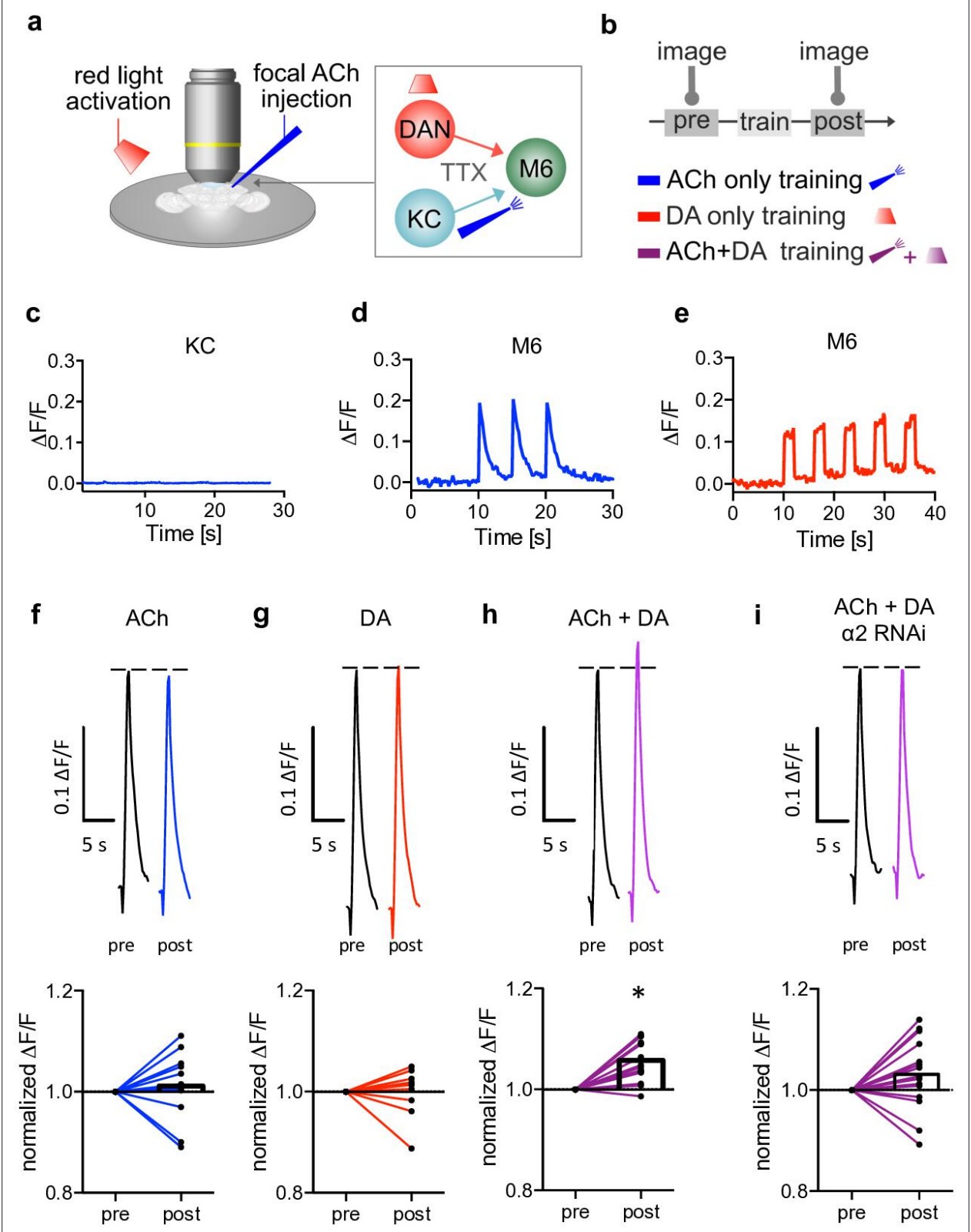

**Figure 3.** Induction of postsynaptic plasticity bypassing the presynapses. (**a**) Left: explant brain experimental configuration. Right: connectivity scheme of mushroom bodies (MB) output synapses. Cholinergic Kenyon cells (KCs) and dopaminergic neurons are presynaptic to M6 MB output neurons (MBONs). Only connections relevant for this protocol are shown for simplicity. Red light pulses trigger release of dopamine (DA) from dopaminergic neurons (R58E02-LexA>lexAop-CsChrimson^tdTomato), while KC input is circumvented and mimicked by focal acetylcholine (ACh; 0.1 mM) injections to

*Figure 3 continued on next page*

*Figure 3 continued*

M6 dendrites in an explant brain preparation. Postsynaptic responses at the level of M6 are measured using GCaMP6f (MB011B [Split-GAL4] >UAS-GCaMP6f). TTX (tetrodotoxine) in the bath suppresses feedback signaling and overall network activity within the circuit. (**b**) Training scheme (top). Baseline responses to ACh application were initially established (pre). Subsequent training protocols consist of either pairing ACh application with simultaneous activation of dopaminergic neurons (purple connection lines), activation of dopaminergic neurons (red light only, red connection lines), or ACh only (blue, connection lines). This is followed by a test trial (post) through ACh application. (**c**) Averaged traces of axonal KC calcium changes induced by focal ACh injections. No apparent transients are observable; n=7. Line is mean ± SEM. (**d**) Sample trace of dendritic M6 calcium changes induced by focal ACh injections. (**e**) Sample trace of dendritic M6 calcium changes induced by red-light pulses. (**f-i**) Above: sample calcium traces in response to ACh injections recorded from M6 dendrites pre (black traces) and post (colored traces) training. Below: peak quantification. (**f**) Changes in ACh-evoked calcium transients; comparison of mean peaks pre- and post-'ACh only' training. Before-after plots and bar graphs (mean); n=13, ratio paired t-test. (**g**) Changes in ACh-evoked calcium transients; comparison of mean peaks pre- and post 'red light only' training. Before-after plots and bar graphs (mean); n=10, ratio paired t-test. (**h**) Changes in ACh-evoked calcium transients; comparison of mean peaks pre- and post 'paired' training. Before-after plots and bar graphs (mean); n=18, ratio paired t-test, *p<0.05. (**i**) Changes in ACh-evoked calcium transients following RNAi knockdown of the α2 subunit in M4/6 MBONs; comparison of mean peaks pre- and post 'paired' training. Before-after plots and bar graphs (mean); n=15, ratio paired t-test. Also see *Figure 3—figure supplement 1* for further information.

The online version of this article includes the following figure supplement(s) for figure 3:

**Figure supplement 1.** Control experiments and non-normalized data display for *Figure 3*.

paired condition (acetylcholine application and red light; *Figure 3h*). This plasticity was not observed when testing after acetylcholine only or dopamine only training (*Figure 3f and g*). Importantly, we also did not observe any changes when pairing acetylcholine application with red light in non-CsChrimson-expressing controls (*Figure 3—figure supplement 1c,d*).

Because we are using global acetylcholine application instead of sparse activation of single synapses, these experiments likely do not reflect in vivo physiological settings (*Owald et al., 2015*). However, our proof of principle experiments demonstrate that postsynaptic plasticity at the level of MBONs can take place independently of the presynapses of the KCs. Intriguingly, we did not observe changes in calcium transient magnitudes when knocking-down α2 (UAS-nAChR$^{RNAi}$) in M4/6 (MB011B-Split Gal4; *Figure 3i*), which is consistent with postsynaptic plasticity being linked to the requirement of nicotinic receptors in memory storage.

## Non-uniform distribution of nAChR α-subunits throughout MB compartments

Our data so far are suggestive of α2-containing nicotinic receptors being involved in appetitive memory storage. To test whether receptor levels were interdependent, we made use of a newly established CRISPR (clustered regularly interspaced short palindromic repeats)-based genomic collection of GFP-tagged endogenous nAChR subunits (for details see Methods) covering all α subunits (with the exception of α3) under control of their endogenous promoter, allowing for analyses of receptor distribution and dynamics in a dense neuropile in situ.

We first characterized receptor subunit signals throughout the 15 MB compartments, several of which have been shown to be involved in the encoding of specific memories. We found a non-uniform distribution (*Figure 4a and b*, *Figure 4—figure supplement 1*) that was unique for each subunit, indicating considerable heterogeneity of receptor composition. α5, which is required for immediate and 3-hr appetitive memories, was abundant throughout the γ lobe, including γ5 (innervated by M6) and slightly less at the level of β'2 (innervated by M4 and in parts by M6). α2 subunits, required for 3-hr appetitive memories, showed similarly high relative abundance in β'2 and γ5 (*Figure 4a and b*). Of note, these subunits were also detected in other MB output compartments, such as α'3, which harbors MBONs involved in non-associative familiarity learning (*Hattori et al., 2017*).

We next evaluated whether the fluorescent signals of the α2 and α5 subunits (with α5 potentially functioning upstream of α2, *Figure 2*) observed in the β'2 and γ5 compartments were derived from receptors within the dendritic processes of M4/6. To do so, we performed cell-specific knockdown experiments using VT1211-Gal4 to drive subunit-specific RNAi (UAS-nAChR$^{RNAi}$) and quantified the relative fluorescent signal of the knockdown compartment relative to the neighboring unmanipulated compartments (*Figure 4c–f*).

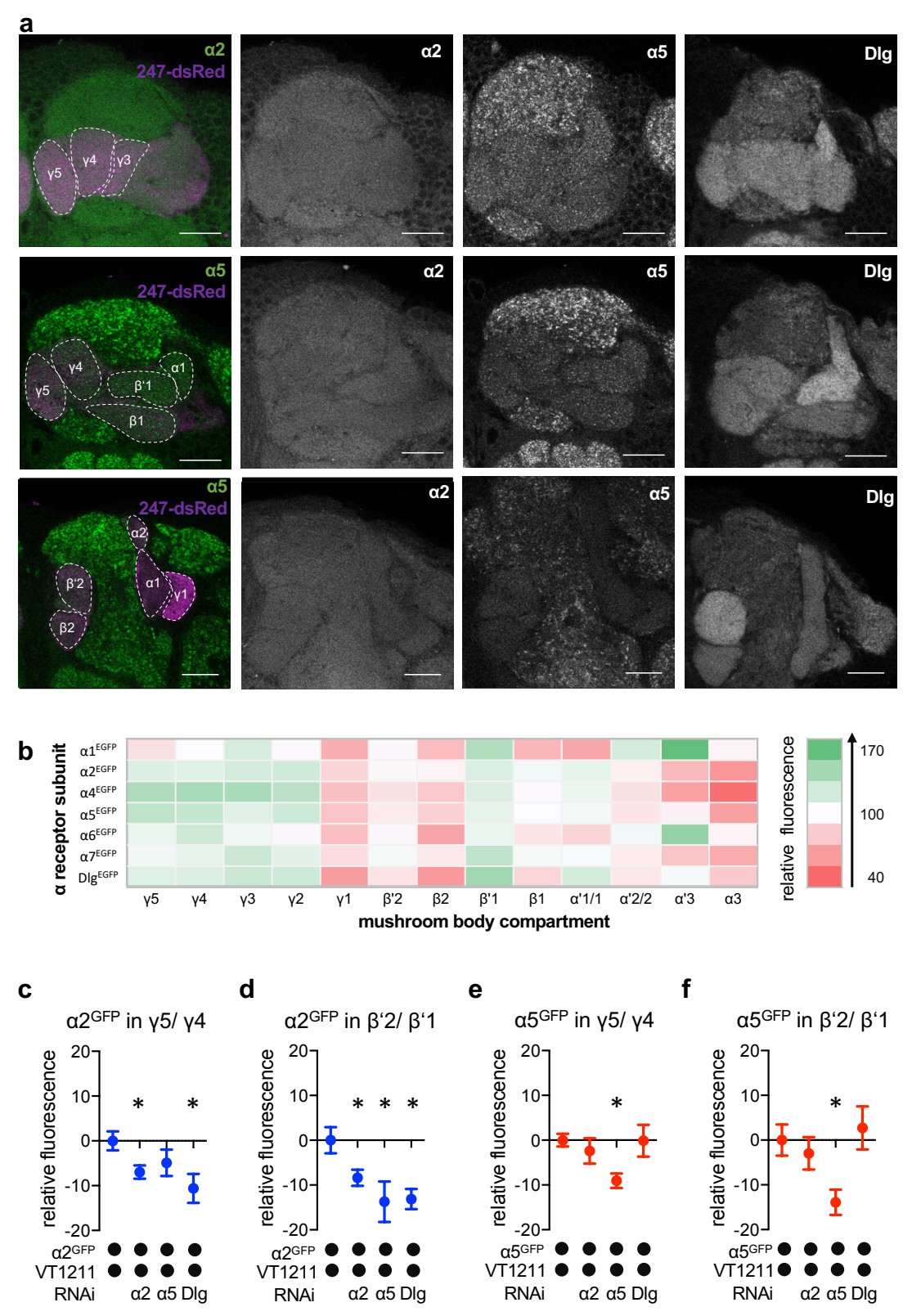

**Figure 4.** Nicotinic acetylcholine receptor (nAChR) α subunit localization throughout the mushroom bodies (MB): MB output neuron (MBON)-specific RNAi alters subunit distribution. (**a**) Representative images of the GFP-tagged nAChR subunits α2 and α5 as well as discs large (Dlg). Individual images displayed here are taken from different animals. For other subunits, see *Figure 4—figure supplement 1*. Scale bar: 20 µm. Left: merge of α subunit signal (green) with MB compartments marked with 247-dsRed (magenta). Compartments are indicated by dashed lines. Top row: γ compartments;

*Figure 4 continued on next page*

*Figure 4 continued*

middle row: α, β′, β, and γ compartments, bottom row: α′, α, and β compartments. (**b**) Quantification of all GFP-tagged α receptors (except for the α3 subunit). GFP signals for the indicated MB compartments are relative to the mean intensity of the GFP signal of the complete MB; n=7–18. (**c**) Knockdown of Dlg or α2 in M4/6 neurons (driver line VT1211-Gal4) significantly reduces the α2$^{GFP}$ fluorescence within the γ5 compartment (relative to unmanipulated γ4). Bar graph: normalized mean ± SEM; n=8–20; Kruskal-Wallis followed by Dunn's test (p<0.05), *p<0.05. (**d**) Knockdown of either the α2 or the α5 nAChR subunit or Dlg in M4/6 neurons (driver line VT1211-Gal4) decreases the relative fluorescence signal of α2$^{GFP}$ in the β′2 compartment (relative to unmanipulated β′1). Bar graph: normalized mean ± SEM; n=9–20; one-way ANOVA followed by Dunnett's test (p<0.05), *p<0.05. (**e**) Knockdown of α5 in M4/6 neurons (driver line VT1211-Gal4) decreases the α5$^{GFP}$ signal in the γ5 compartment (relative to unmanipulated γ4). Bar graph: normalized mean ± SEM; n=9–28; Kruskal-Wallis followed by Dunn's test (p<0.05), *p<0.05. (**f**) α5$^{GFP}$ fluorescence is significantly decreased in the β′2 compartment (relative to unmanipulated β′1) after knockdown of α5 in M4/6 neurons (driver line VT1211-Gal4). Bar graph: normalized mean ± SEM; n=10–28; Kruskal-Wallis followed by Dunn's test (p<0.05), *p<0.05. Also see *Figure 4—figure supplement 1* for further information.

The online version of this article includes the following figure supplement(s) for figure 4:

**Figure supplement 1.** Detailed distribution of α subunits in the mushroom bodies (MB) accompanying *Figure 4*.

Knockdown of the α5 nAChR subunit reduced the relative α5$^{GFP}$ signal specifically and significantly in γ5 and β′2 (*Figure 4e and f*). α5 abundance was, however, unaltered when knocking down α2 or Dlg, which is in line with α5 functioning as a possible trigger for plasticity processes.

Likewise, confirming that the observed signal was derived from M4/6 MBON dendrites, α2 knockdown reduced relative α2$^{GFP}$ levels significantly throughout the β′2 and γ5 compartments (*Figure 4c and d*). Strikingly, we also observed reduced α2 nAChR subunit levels following α5 subunit knockdown in the β′2 compartment or Dlg knockdown in the β′2 and γ5 compartments (*Figure 4c and d*), which would be in line with a Dlg-dependent sequential requirement of receptor subunits during memory formation (also compare behavioral data in *Figure 2*).

Our data therefore are consistent with a role of α5 nAChR subunits and Dlg functioning upstream of α2 subunit-positive receptors, at least within the β′2 compartment.

## nAChR subunits shape synaptic MB output properties

We next focused on implications of α2 subunit knockdown on postsynaptic function of M4/6 MBONs. Axonal calcium transients had previously been shown to be decreased following knockdown of α subunits (*Barnstedt et al., 2016*). However, depending on the overall topology of dendritic input sites, both increased or decreased postsynaptic drive could lead to changed dendritic integration properties or potential interference of synaptic inputs, resulting in reduced signal propagation (*Stuart and Spruston, 2015*).

To directly test dendritic responses, we expressed UAS-GCaMP6f in M4/6 MBONs (VT1211-Gal4) (*Figure 5a*) and exposed the flies repeatedly to alternating puffs of the odors octanol (OCT) and methylcyclohexanol (MCH) (*Figure 5b*). We focused our experiments on the β′2 compartment (*Figure 5*), as this is innervated by M4 MBONs that show input-specific plasticity following appetitive learning (*Owald et al., 2015*). Initial dendritic odor responses were not different between α2 subunit knockdown (VT1211-Gal4>UAS-α2$^{RNAi}$) and controls (*Figure 5c and d*), while initial odor-evoked dendritic calcium transients were elevated following knockdown of α5 (*Figure 5c, d* and *Figure 5—figure supplement 1b, c*). Neither control animals nor α2 knockdown animals (VT1211-Gal4>UAS-α2$^{RNAi}$) showed significant changes in odor-specific calcium transients after several exposures to MCH (*Figure 5e and f*). Odor responses following α5 knockdown (VT1211-Gal4>UAS-α5$^{RNAi}$), however, clearly depressed after multiple odor exposures (*Figure 5g*), indicating that loss of α5, in comparison to the controls, can actually lead to synapses being potentiated (synaptic weights are already high) from the start, even prior to the application of odors. On the contrary, compared to controls, α2 nAChR subunit knockdown interfered with odor-evoked transmission to a lesser extent. We did not observe any changes in calcium signals at the level of the corresponding KC axons, further supporting that the observed plasticity was of postsynaptic origin (*Figure 5—figure supplement 1g,h*).

## The α2 nAChR subunit is required for the formation of appetitive memory traces in vivo

Of note, we observed a facilitation in M4/6 of controls when using repeated application of OCT, indicating that some parameters underlying the observed responses are odor-dependent (*Figure 5—figure supplement 1d*). However, the only slight effects on MCH-induced responses observed for α2

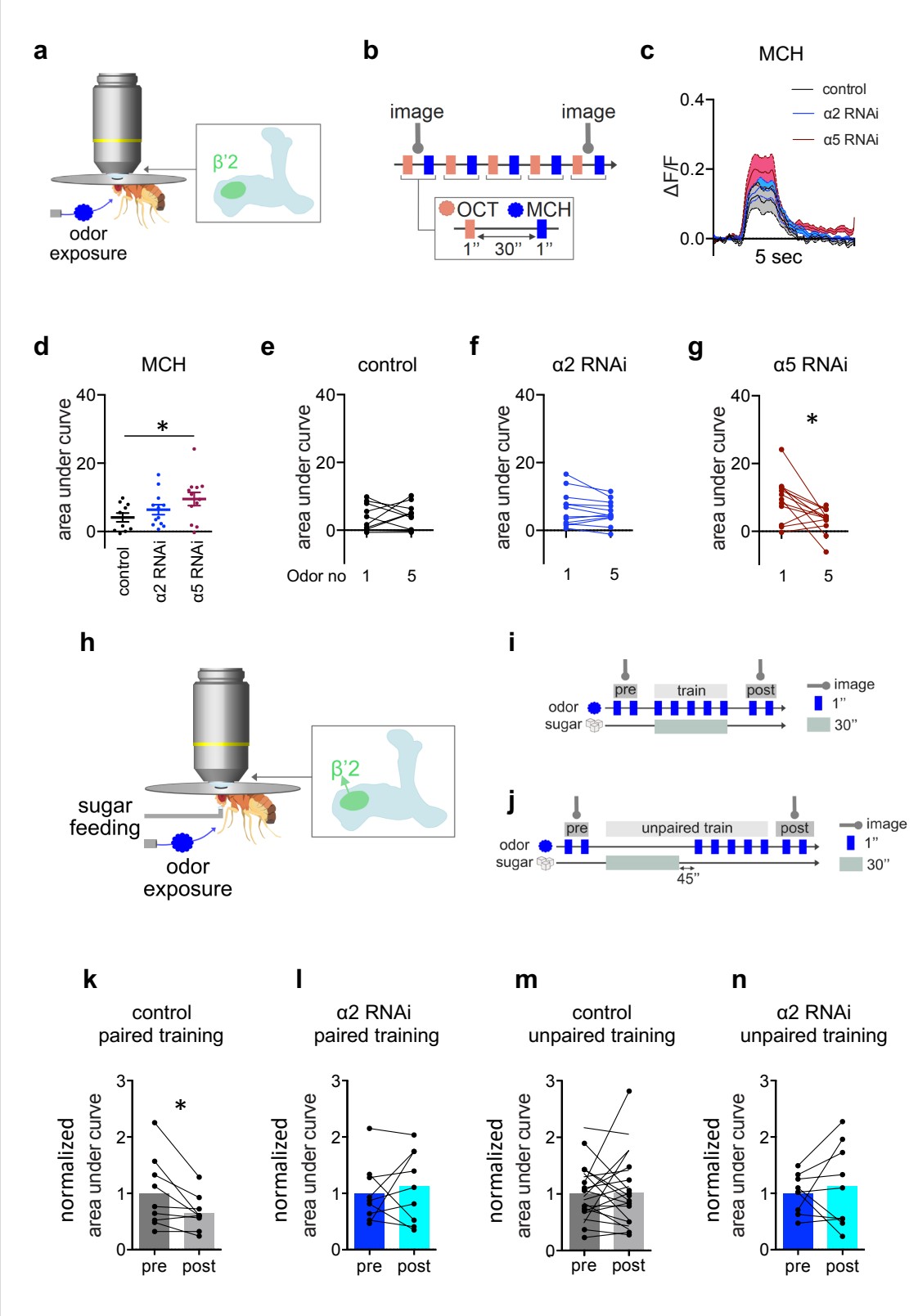

**Figure 5.** α2 is required for learning-associated plasticity in vivo. (**a**) Scheme indicating the dendritic imaging area (**c–g**) at the level of the β'2 compartment. (**b**) Odor exposure protocol. Five octanol (OCT) stimuli were alternatingly administered with five methylcyclohexanol (MCH) stimuli. 1-s odor puffs were separated by 30 s breaks. (**c**) Averaged traces of GCaMP6f (calcium) responses to MCH from control (black), α2 (blue), and α5 RNAi (red; driven in M4/6, respectively, driver line VT1211-Gal4) flies. Solid traces are mean, shaded areas SEM; n=10–12. (**d**) Area under curve (AUC) quantification

*Figure 5 continued on next page*

*Figure 5 continued*

of averaged initial odor responses to MCH following α5 knockdown in M4/6 neurons (driver line VT1211-Gal4). Mean ± SEM; one-way ANOVA followed by Dunnett's test; *$p<0.05$; n=10–12. (**e**) Control flies show no significant increase between the first and the fifth response to MCH. n=10; paired t-test. (**f**) α2 RNAi flies show no difference between the first and fifth odor response to MCH. Nicotinic acetylcholine receptor (nAChR) subunit RNAi is driven in M4/6 neurons (driver line VT1211-Gal4). n=12; paired t- test. (**g**) α5 RNAi flies show a significant decrease in calcium transients over the course of consecutive odor exposures. nAChR subunit RNAi is driven in M4/6 neurons (driver line VT1211-Gal4). n=12; Wilcoxon matched-pairs signed rank test; *$p<0.05$. (**h**) Arrow in scheme for absolute training indicates the axonal imaging area of M4 (**k–n**). (**i**) Scheme for absolute paired training under the microscope: flies are exposed to two brief stimuli of the odor (MCH) before training while recording GCaMP6f signals (preresponse). Immediately after, flies are presented with a sugar solution for 30 s accompanied by 5 pulses of odor stimuli. 1 min after training, flies are exposed to two brief odor stimuli again while recording GCaMP6f signals (postresponse). Axonal imaging area indicated by arrow. (**j**) Scheme for absolute unpaired training under the microscope: as in (**i**) except flies are exposed to 5 pulses of odor stimuli 45 s after presenting the sugar solution for 30 s. 1 min after training, the flies are exposed to two brief odor stimuli again while recording GCaMP6f signals (postresponse). Axonal imaging area indicated by arrow. (**k**) Control flies show a significant decrease in odor-evoked GCaMP6f signals. AUC of response to MCH (trained odor) following paired training (driver line VT1211-Gal4, axonal imaging). Before-after plots and bar graphs (mean); n=9; paired t-test; *$p<0.05$. (**l**) α2 RNAi flies show no significant decrease in odor-evoked GCaMP6f signals. AUC of response to MCH (trained odor) following paired training (driver line VT1211-Gal4, axonal imaging). Before-after plots and bar graphs (mean); n=9; paired t-test. (**m**) Control flies show no significant decrease in odor-evoked GCaMP6f signals. AUC of response to MCH (trained odor) following unpaired training (driver line VT1211-Gal4, axonal imaging). Before-after plots and bar graphs (mean); n=12; Wilcoxon matched-pairs signed rank test. (**n**) α2 RNAi flies show no significant decrease in odor-evoked GCaMP6f signals. AUC of response to MCH (trained odor) following unpaired training (driver line VT1211-Gal4, axonal imaging). Before-after plots and bar graphs (mean); n=9; paired t-test. Also see *Figure 5—figure supplement 1* for further information.

The online version of this article includes the following figure supplement(s) for figure 5:

**Figure supplement 1.** Additional data and display of non-normalized data accompanying *Figure 5*.

knockdown animals, allowed us to next investigate the role of α2 during in vivo associative appetitive learning.

We performed in vivo training under the microscope experiments using an absolute paradigm, pairing odor (MCH) exposure with sugar feeding during training, this time imaging from the axonal compartment to assess the integrated signal originating from the dendritic input. Comparing odor responses at the level of M4 (VT1211-Gal4>UAS-GCaMP6f) before and after training revealed a significant depression for control animals in line with previous observations (*Owald et al., 2015*; *Lewis et al., 2015*; *Figure 5k*, *Figure 5—figure supplement 1l*). Importantly, this training-induced depression was neither observed in controls, where odor and sugar were administered unpaired (*Figure 5m*, *Figure 5—figure supplement 1n*), nor after MBON-specific knockdown of α2 (VT1211-Gal4>UAS-GCaMP6f, UAS-α2^RNAi; *Figure 5l and n*, *Figure 5—figure supplement 1m,o*).

Together, our data point toward a mechanism, where nicotinic receptor subunits shape synaptic properties (*Figure 5*), with α2 as a postsynaptic substrate underlying appetitive training-induced plasticity processes.

## In vivo imaging of postsynaptic receptor plasticity reveals altered α2 dynamics

Structural changes at the level of the receptor composition are hallmarks of postsynaptic plasticity expression in vertebrates. Typically, rearrangements can be measured by altered dynamics (or motility) of receptors that can reflect incorporation or removal of receptor complexes. We next sought to test whether dynamic receptor behavior could serve as a structural correlate of cholinergic postsynaptic memory trace expression. To do so, we turned to in vivo imaging experiments of the endogenously tagged α2 subunit (α2^GFP; *Figure 4*, *Figure 6*, *Figure 6—figure supplement 1*). Following artificial appetitive training protocols (*Figure 6a–c*, focal dopamine only, odor only, or odor paired with focal dopamine, consecutively, see methods), in situ receptor dynamics were estimated at the level of the β'2 compartment of the MB by measuring fluorescence recovery after photobleaching (FRAP; *Figure 6a–c*).

Exposing the fly to MCH induced significantly increased fluorescence recovery when compared to dopamine injections only or odor paired with dopamine (*Figure 6d and e*). Dopamine, therefore, does not induce plasticity on its own, and furthermore, it suppresses odor-induced recovery when applied simultaneously with an odor. To control for recovery either depending on the type of odor used or the order of conditions applied, we next conducted similar experiments using OCT this time with each condition applied in separate flies. We only observed significant recovery in the odor only condition,

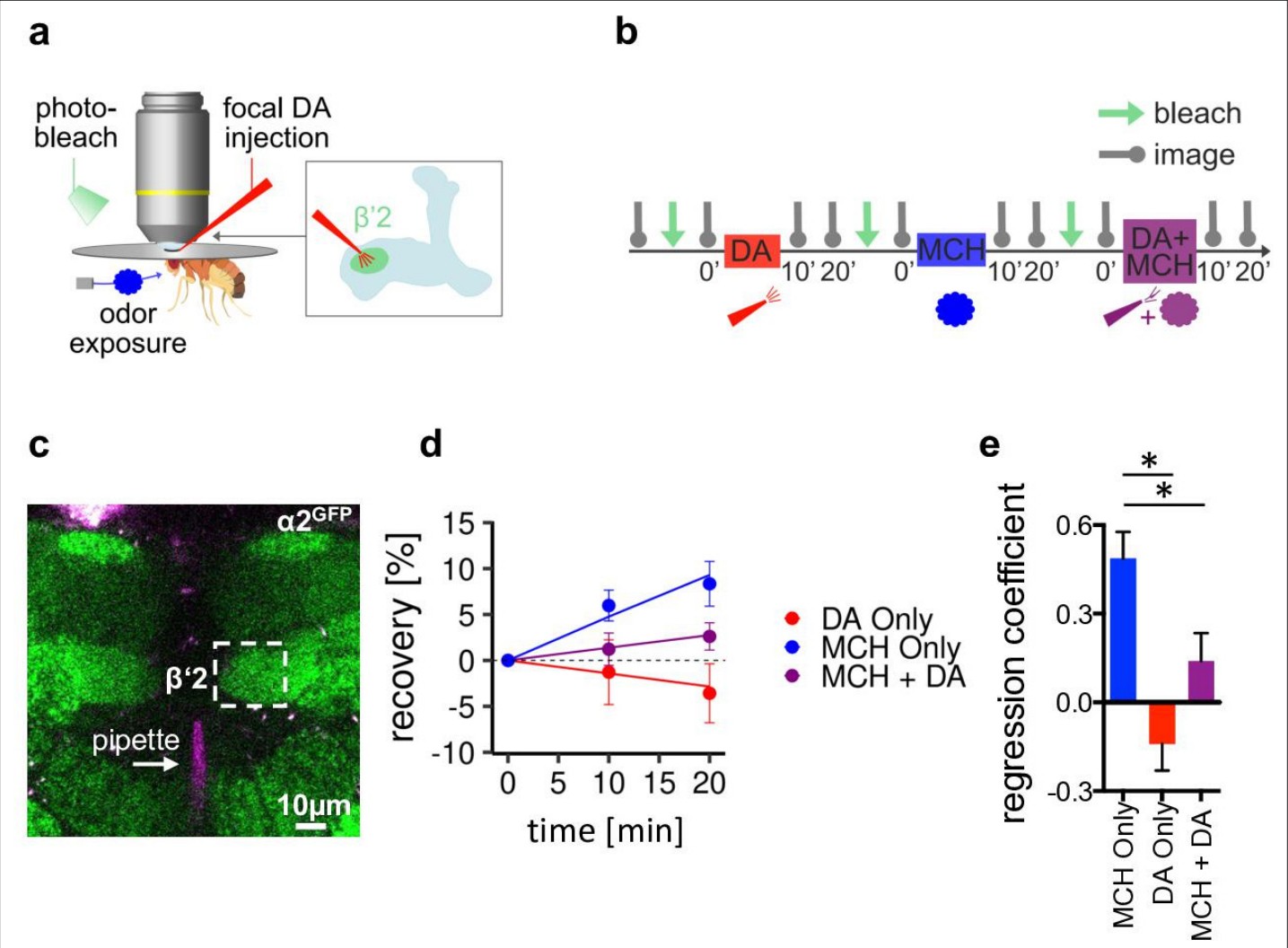

**Figure 6.** α2 nicotinic acetylcholine receptor (nAChR) subunits dynamically rearrange. (**a**) In vivo imaging configuration and scheme of site of dopamine (DA) injection during fluorescence recovery after photobleaching (FRAP) experiments at the level of the Kenyon cell to mushroom body output neuron synapses of the β′2 compartment. (**b**) FRAP experimental protocol. After bleaching, the baseline picture was taken followed by DA injection, odor presentation, and odor presentation simultaneously with DA injection in the same fly. Fluorescence recovery was monitored at the 10- and 20-min time points. (**c**) Example image of α2$^{GFP}$ expression; white dashed box shows the β′2 output zone; DA injection pipette (with Texas red) is labeled in magenta. Scale bar: 10 μm. (**d**) Inverse exponential decay fit of FRAP following methylcyclohexanol (MCH) exposure (blue line), MCH exposure simultaneously with DA injection (purple line), and DA injection alone (red line). (**e**) Regression coefficient for the inverse exponential decay fit. Bar graphs: regression coefficients of recovery kinetics ± standard error of regression; n=9–10, linear mixed effects model followed by pairwise comparison from estimated marginal trends. *p < 0.05. Also see *Figure 6—figure supplement 1* for further experiments.

The online version of this article includes the following figure supplement(s) for figure 6:

**Figure supplement 1.** Receptor subunit recovery, accompanying *Figure 6*.

whereas dopamine and odor paired with dopamine induced no significant recovery (*Figure 6—figure supplement 1a-d*).

Thus, our data indicate that pairing odor presentation with dopamine application stalls α2$^{GFP}$ dynamics, potentially by either stabilizing the already present amount of receptor or hindering new incorporation of α2-containing receptors. Interestingly the opposite, increased receptor dynamics, is observed after odor exposure without reinforcer. Note that the absence of acute stimuli (constant air stream only) did not induce signal recovery, demonstrating that it is the presence of the odor that changes baseline receptor behavior (*Figure 6—figure supplement 1e*). Thus, stalling α2 dynamics

can be correlated to depression of M4/6 MBON synapses *Owald et al., 2015*; *Felsenberg et al., 2018* following appetitive training (*Figure 5k*).

## Familiarity learning alters postsynaptic receptor dynamics

Our data so far suggest that regulation of α2 subunits downstream of α5 is involved in postsynaptic plasticity mechanisms underlying appetitive but not aversive memory storage. Besides associative memories, non-associative memories, such as familiarity learning, a form of habituation, are also stored at the level of the *Drosophila* MBs. We next asked whether postsynaptic plasticity expressed through α5 and α2 subunit interplay, was exclusive to appetitive memory storage, or would represent a more generalizable mechanism that could underlie other forms of learning represented in the MBs. We turned to the α'3 compartment at the tip of the vertical MB lobe that has previously been shown to mediate odor familiarity learning. This form of learning allows the animal to adapt its behavioral responses to new odors and permits for assaying direct odor-related plasticity at the level of a higher-order integration center. Importantly, this compartment follows different plasticity rules, because the odor serves as both the conditioned (activating KCs) and unconditioned stimulus (activating corresponding DANs; *Hattori et al., 2017*). While allowing us to test whether the so far uncovered principles could also be relevant in a different context, it also provides a less complex test bed to further investigate whether α5 functions upstream of α2 dynamics.

Confirming previous observations (*Hattori et al., 2017*), a repeated odor application paradigm (*Figure 7a*) led to the depression of postsynaptic calcium transients at the level of the α'3 MBONs (MB027B Split-Gal4 >UAS-GCaMP6f; *Figure 7b*). We did not detect a corresponding depression on the presynaptic side when imaging arbors of a sparse α'β' KC driver line (MB369B Split-Gal4 >UAS-GCaMP6f) within α'3 (*Figure 7b*), further indicating that memories were predominantly stored postsynaptically in this compartment. We next performed in vivo FRAP experiments following familiarity learning paradigms. After odor training, we observed clear recovery rates of α2$^{GFP}$ signals compared to the control group, however, not of α5$^{GFP}$ or Dlg$^{GFP}$ (*Figure 7c–f*, *Figure 7—figure supplement 1*). Therefore, increased α2 subunit dynamics are triggered through training events and, at the level of the α'3 compartment, accompany postsynaptic depression of the MBONs.

To invariantly test whether the observed recovery was attributable to α2 expressed in α'3 MBONs, we trained animals while knocking down α2 specifically in α'3 MBONs (α2$^{GFP}$ and MB027B Split-Gal4 >UAS-α2$^{RNAi}$). In accordance with the observed signal recovery deriving from MBONs, no recovery was observed after α2 knockdown (*Figure 7g*). Importantly, we did not observe α2$^{GFP}$ recovery when performing specific α5 knockdown in α'3 MBONs (α5$^{GFP}$ + MB027B Split-Gal4 >UAS-α5$^{RNAi}$, *Figure 7g*), indicating a role of α5 upstream of α2 also in this compartment.

## α5 subunits govern induction and α2 subunits expression of non-associative familiarity learning

Finally, we tested whether interfering with α5 and α2 nAChR subunits at the level of α'3 MBONs (MB027B Split-GAL4) would also impact familiarity learning behavior (*Figure 8*). Flies were covered in dust and subjected to repeated odor exposures (*Hattori et al., 2017*). As expected (*Hattori et al., 2017*), control flies readily groomed to remove the dust, however, typically stopped this action when detecting the novel odor (*Figure 8a–c*). Over subsequent trials, control flies learned that this odor was familiar and stopped reacting to the stimulus, continuing grooming (*Figure 8a–c*, *Figure 8—figure supplement 1*). Expressing RNAi to the α2 subunit (UAS-nAChR$^{RNAi}$) at the level of the α'3 MBONs (MB027B Split-Gal4) clearly impacted learning: flies learned with decreased efficacy and only after several trials (*Figure 8a–d*). Strikingly, α5 RNAi-expressing flies failed to stop grooming even to the first stimulus. Indeed, they acted as if they had already learned that an odor was familiar (*Figure 8a–d*).

Together, our data are in line with a model where α5 can induce memory formation, while lack of α5 leads to fully potentiated synapses. Subsequent expression of memory traces requires α2-containing receptors. Importantly, recovery accompanies synaptic depression at the level of the α'3 MBONs, while being suppressed by paired training in the β'2 compartment. Moreover, α2 appears to be involved in both depression and facilitation of synapses. Thus, synapses could bidirectionally utilize plasticity of the same receptor subunit for storing different types of information (*Figure 9*).

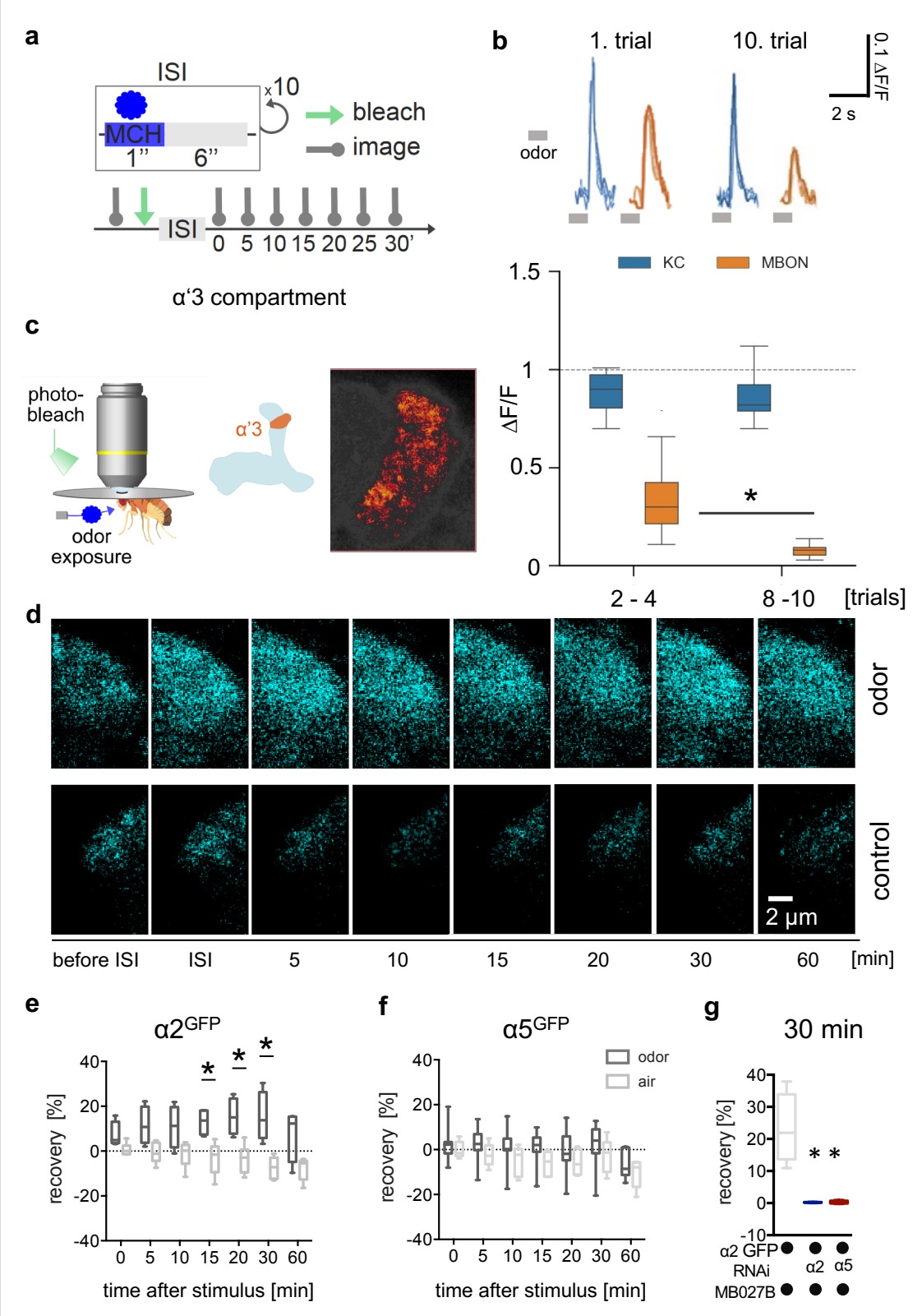

**Figure 7.** Non-associative plasticity alters postsynaptic α2 subunit receptor dynamics. (**a**) Training scheme indicating odor application, bleaching, and imaging time points. MCH was given 10 times for 1 s with a pause of 6 s in-between. Images were taken after training in absence of odor immediately afterward and 5, 10, 15, 20, 30, and 60 min later. (**b**) Top: Calcium peaks in response to odor stimuli of presynaptic Kenyon cells (KCs; MB369B as driver line) and adjacent postsynaptic mushroom body output neurons (MBONs; driver line: MB027B [Split-GAL4]). Individual calcium responses to trials

*Figure 7 continued on next page*

Figure 7 continued

1 and 10 for MBONs (orange lines) and KCs (blue lines). Bottom: Averaged calcium responses to odor stimuli of presynaptic KCs and postsynaptic MBONs of trials 2–4 and 8–10, respectively. Responses decrease at the level of MBONs but not at the level of KCs over 10 trials. Box plots are median and 75% quartiles; n=15; Kruskal-Wallis followed by Dunn's test (p<0.05), *p<0.05. (c) In vivo imaging configuration (left), scheme of α'3 compartment analyzed (right), and representative α5$^{GFP}$ fluorescent image (smoothed). Scale bar: 10 µm. (d) Example images of α2$^{GFP}$ fluorescence recovery after photobleaching (FRAP) experiment at the level of the α'3 compartment at specific time points before and after training. Top row: after training; bottom row: control settings. Scale bar: 2 µm. (e) FRAP of α2$^{GFP}$ nicotinic acetylcholine receptor (nAChR) subunit in the α'3 compartment after odor presentation. α2$^{GFP}$ shows significant recovery following odor training compared to the controls. Recovery rate is normalized to the baseline recorded after selective bleaching of the α'3 mushroom bodies (MB) compartment. Box plots are median and 75% quartiles; n=4–6; multiple t-tests with Sidak-Bonferroni correction, *p<0.05. (f) FRAP of α5$^{GFP}$ subunit in the α'3 compartment after odor presentation. α5$^{GFP}$ did not show significant recovery compared to the controls. Recovery rate is normalized to the baseline recorded after selective bleaching of α'3 MB compartment. Box plots are median and 75% quartiles; n=5–7, multiple t-tests with Sidak-Bonferroni correction. (g) FRAP of α2$^{GFP}$ nAChR subunit in the α'3 compartment after odor presentation and knockdown of either the α2 or α5 subunit (RNAi) in the α'3 MBON (driver line MB027B [Split-GAL4]). α2$^{GFP}$ shows no recovery 30 min after odor training in α2 or α5 knockdown animals compared to the controls. Recovery rate is normalized to the baseline recorded after selective bleaching of the α'3 MB compartment. Box plots are minimum value to maximum value; n=4–5; Kruskal-Wallis followed by Dunn's test (p<0.05), *p<0.05. Also see *Figure 7— figure supplement 1* for further information.

The online version of this article includes the following figure supplement(s) for figure 7:

**Figure supplement 1.** Discs large GFP (Dlg$^{GFP}$) fluorescence recovery after photobleaching (FRAP), accompanying *Figure 7*.

## Discussion

Synaptic plasticity that manifests itself in synaptic weight changes is widely recognized as substrate for memory storage throughout the animal kingdom. How synapses adapt in order to change their efficacy during learning has been a focus of attention over the last decades. While it is undisputed that both pre- and postsynaptic mechanisms of memory storage exist in vertebrates, invertebrate memory-related synaptic plasticity has been largely localized to the presynaptic compartment (*Bilz et al., 2020*; *Boto et al., 2014*; *Handler et al., 2019*; *Ehmann et al., 2018*; *Tully et al., 1994*). However, the core of the debate boils down to a key question: do vertebrates and invertebrates use similar mechanisms to store memories or are there fundamental differences? A first clear difference appears to be the use of different neurotransmitter systems, glutamate and acetylcholine, respectively, in the vertebrate and *Drosophila* learning centers (*Barnstedt et al., 2016*).

### Postsynaptic plasticity in associative memory storage

Here, we use the genetic tractability of the *Drosophila* system to directly address postsynaptic plasticity during olfactory memory storage in invertebrates. Large amounts of evidence from *Drosophila* so far suggest a presynaptic mode of memory storage (*Bilz et al., 2020*; *Boto et al., 2014*; *Handler et al., 2019*; *Ehmann et al., 2018*; *Tully et al., 1994*). Moreover, several studies indicated that block of KCs during learning does not interfere with memory performance (*Dubnau et al., 2001*; *McGuire et al., 2001*; *Schwaerzel et al., 2002*). However, other studies blocking KC subsets did find impairments (*Krashes et al., 2007*; *Yamazaki et al., 2018*; *Trannoy et al., 2011*) in the context of short-term appetitive memory, while downstream circuit elements have been implicated in appropriate memory formation (*Pai et al., 2013*; *Widmer et al., 2018*; *Wu et al., 2017*). Here, we revisited such experiments and found, in accordance with previous studies, only mild, if any, requirement for aversive memory storage. We, however, fully abolished appetitive memories (*Figure 1*) by blocking KC output during acquisition, providing the basis for a model of postsynaptic plasticity (*Figure 9*) that is induced and expressed through distinct nAChR subunits (*Figures 2–8*).

Our study hints toward different pre- and postsynaptic storage mechanisms underlying aversive and appetitive memories. It also argues against the assumption that appetitive and aversive memories will necessarily use the same molecular machinery to store information. Interestingly, arguing for a division of appetitive and aversive storage sites, subpopulations of KCs have been implicated in aversive and appetitive memory, respectively (*Perisse et al., 2013*). Moreover, postsynaptic contributions were previously ruled out for a synaptic junction required for storage of aversive but not appetitive memories, which is fully consistent with our findings (*Hige et al., 2015*). However, a recent study investigating postsynaptic calcium transients across MB compartments could be in line with postsynaptic modifications occurring following aversive training in some MB output compartments (*Hancock et al., 2022*). Additionally, the requirement of MBON signaling has been demonstrated, particularly

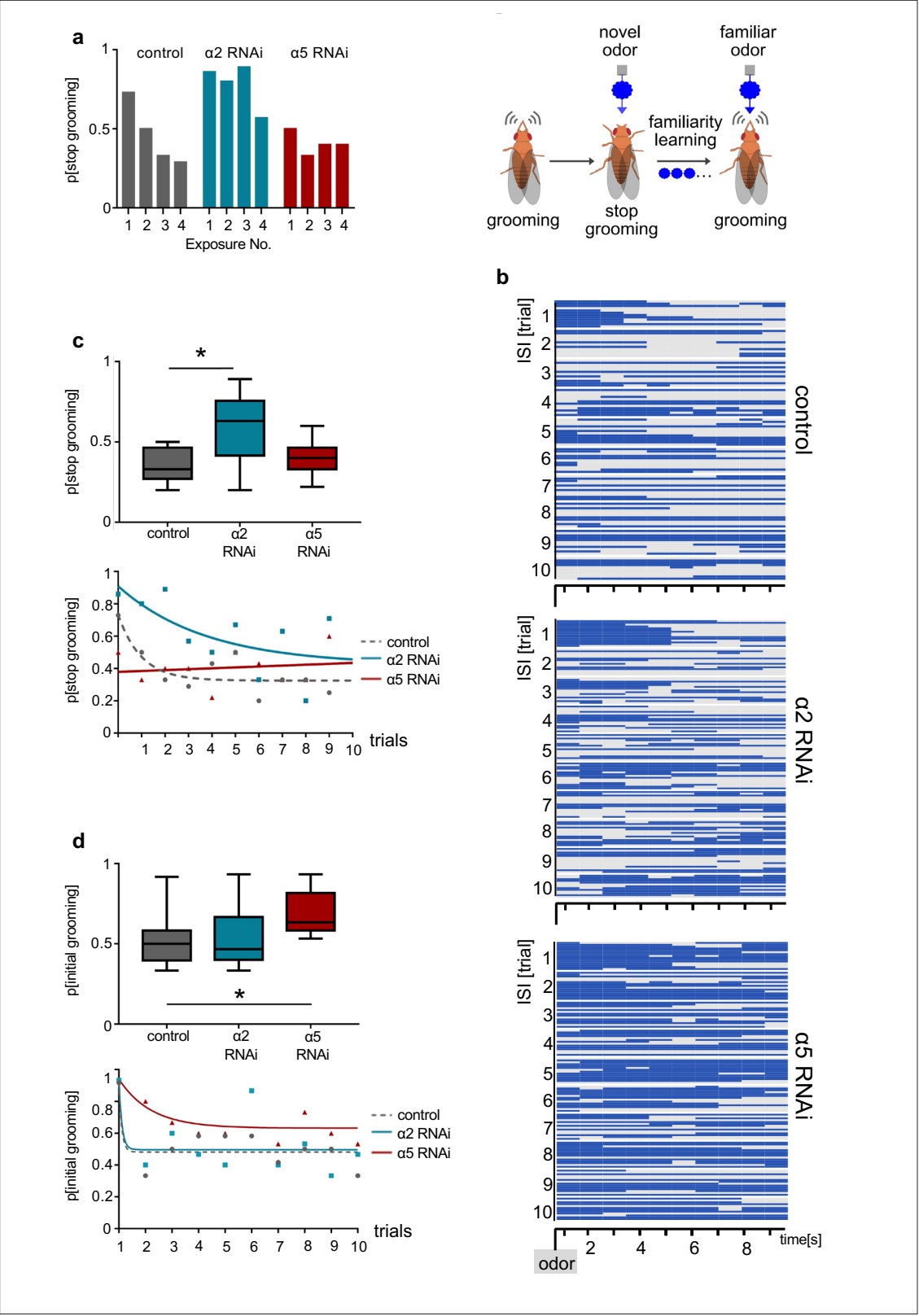

**Figure 8.** α2 and α5 nicotinic acetylcholine receptor (nAChR) subunits are required for non-associative familiarity learning at the level of α′3 mushroom body output neurons (MBONs). (**a**) Scheme of behavioral responses to novel and familiar odors (right). (Left): Knockdown of α nAChR subunits at the level of α′3 MBONs alters odor familiarity learning and the probability to stop grooming. α2 RNAi knockdown delays familiarity learning effects to novel odors. α5 RNAi knockdown flies do not show a novelty response at all. (**b**) Grooming behavior response of dusted flies following the repeated

*Figure 8 continued on next page*

*Figure 8 continued*

presentations of a novel odor (MCH). Ethogram of grooming behavior (blue) during 10 intervals of odor exposures. Horizontal lines in each trial correspond to a single experimental fly within a trial group. Not grooming (gray) flies can further be categorized between pausing and wandering (see *Figure 8—figure supplement 1*). n=15. (**c**) Knockdown of α2 subunit in α'3 MBONs (driver line MB027B [Split-GAL4]) impairs odor familiarity learning significantly; animals show a higher probability to terminate grooming responses during the learning period. The learning period is defined as the odor exposure rounds following the first exposure. Bottom graph: Non-linear representation of grooming flies over 10 training trials. Note that α5 behavioral responses are best described by linear representation. Box plots are median and 75% quartiles; n=9, one-way ANOVA followed by Dunnett's test (p<0.05) *p<0.05. (**d**) Knockdown of α5 subunits in α'3 MBONs (driver line MB027B [Split-GAL4]) leads to an increased probability to start grooming earlier. Bottom graph, non-linear representation of grooming flies over 10 training trials. Box plots are median and 75% quartiles; n=9, Kruskal-Wallis followed by Dunn's test (p<0.05), *p<0.05. Also see *Figure 8—figure supplement 1* for further information.

The online version of this article includes the following figure supplement(s) for figure 8:

**Figure supplement 1.** Additional ethograms, accompanying *Figure 8*.

in the context of longer-term memory storage (*Pai et al., 2013*; *Ichinose et al., 2021*; *Widmer et al., 2018*; *Wu et al., 2017*). Thus, we do not wish to exclude a potential involvement of postsynaptic plasticity in aversive memory formation per se. On the contrary, it is conceivable that aversive memories also could have an appetitive component (release from punishment).

## Postsynaptic plasticity at the KC to MBON synapse

Recent anatomical studies *Takemura et al., 2017*; *Eichler et al., 2017* have reported both dopaminergic innervation of presynaptic KC compartments and, somewhat unexpectedly, direct synapses between presynaptic dopaminergic terminals and MBONs. We devised an experiment where we substituted KC input to the postsynaptic MBON compartment through artificial acetylcholine injection, while rendering DANs switchable through optogenetics. A protocol that trained and subsequently tested the synaptic junction between KCs and MBONs, demonstrates that plasticity (represented by a change in calcium responses to acetylcholine injection) was inducible by pairing dopaminergic with postsynaptic MBON activation that lasted beyond the training stage and was observable by mere 'recall-like' activation of the system (*Figure 3*).

Our proof-of-principle experiments uncovered the ability to potentiate after pairing M6 MBON activation and stimulating a broad population of DANs that convey information on sugar, water, or the relative valence of aversive stimuli (*Owald and Waddell, 2015*), while we find postsynaptic plasticity to be required for appetitive memory performance (*Figure 3*). However, previous studies looking into 'natural' appetitive sugar conditioning uncovered a relative depression in M4 (another MBON of the M4/6 cluster) dendrites, when comparing the responses of the paired (CS+) and unpaired odor (CS−) 1 hr after appetitive conditioning (*Owald and Waddell, 2015*; *Owald et al., 2015*; *Felsenberg et al., 2018*). Moreover, we here show that in vivo appetitive absolute training depresses subsequent responses to the trained odor (*Figure 5*). It is important to note that, here (*Figure 3*), for our in vitro experiments, we perform global activation of the postsynaptic compartment and not the natural typical coverage of 5% of input synapses per odor (*Honegger et al., 2011*) (that allow for differential conditioning). Induced changes are therefore likely not comparable to the natural settings, where sparse sets of KCs and DANs are active within a tight temporal window. Moreover, we here abolish network contributions (by suppressing active signal propagation), to be able to concentrate on synaptic mechanisms during plasticity induction. Thus, our artificial training (*Figure 3*) through global dendritic activation likely does not mirror precise physiological conditions, allowing for plasticity of a sparse set of synapses to convey odor specificity to a memory. However, the relatively small amplitude of plasticity (here: potentiation) observed actually fits previous (*Owald et al., 2015*) in vivo observations. Moreover, similar protocols (*Zhao et al., 2018*) that involved broad activation of KCs (and thus did not circumvent the presynaptic compartment) have demonstrated comparable plasticity induction at this synaptic junction. It should also be noted that because we are using TTX and local training of KC to M6 synapses in our experiments, we are furthermore missing additional disinhibition that in vivo is mediated via the GABAergic MVP2 MBON (*Felsenberg et al., 2018*; *Perisse et al., 2016*). Of note, how difficult it can be to infer how dendrites compute and integrate all input channels is exemplified by the observation that high levels of odor-mediated dendritic activation after α5 knockdown (*Figure 5*, *Figure 5—figure supplement 1*) appear to be translated to reduced axonal calcium transients (*Barnstedt et al., 2016*), effectively leading to decreased signal transduction within the MBON.

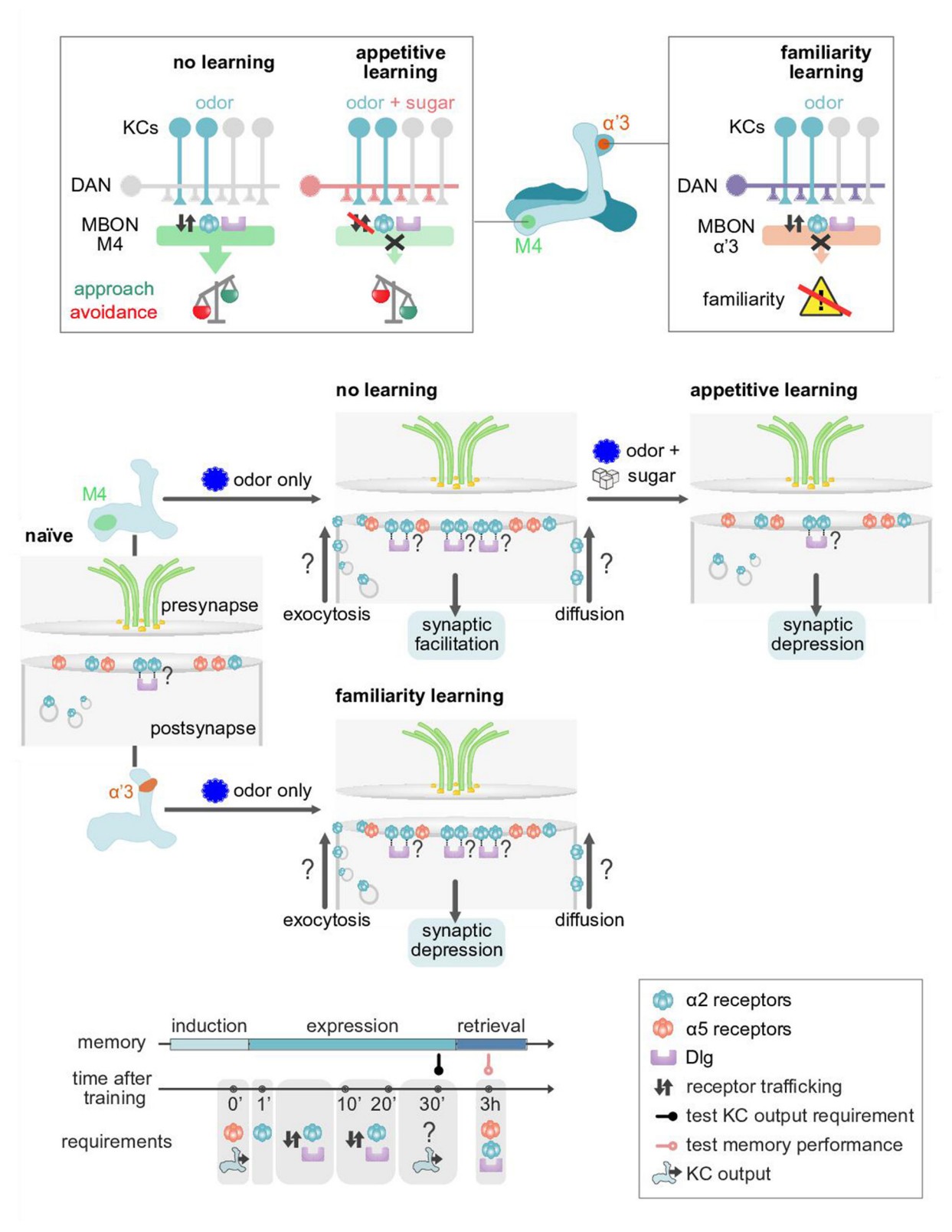

**Figure 9.** Model of postsynaptic plasticity sequence across compartments. Top panels (circuit and behavior level): Mushroom body (MB) compartments investigated. Odors elicit high responses in M4 neurons inducing α2 receptor dynamics. High activity in M4 tilts the balance toward odor avoidance. Learning (or concurrent odor exposure with dopamine application) reduces receptor exchange. This results in reduced postsynaptic responses and tilts the balance toward approach behavior. Suppressed dynamics through learning could be reminiscent of a 'dark current' mechanism as found in

*Figure 9 continued on next page*

*Figure 9 continued*

the mammalian visual system (please see Discussion). At the level of the α′3 compartment repeated odor exposure triggers increased α2 receptor dynamics. In this compartment, the increased dynamics result in reduced odor-evoked activity in the MB output neurons (MBONs) resulting in familiarity learning (less behavioral responsiveness to familiar compared to novel odors). Middle panels (synaptic level): Our data are consistent with a model in which α5 subunit containing receptors (red) mediate the early phase of postsynaptic memory storage, potentially by leading to elevated calcium flux (not addressed in this study) at individual postsynaptic densities (see Discussion and *Figure 10* showing separated PSDs (postsynaptic densities) and analyses concerning input specificity). Concurrent events see changed dynamics of the α2 receptor (blue). Changed dynamics likely reflect changed exocytosis of intracellular receptor populations or lateral diffusion across membranes. Nicotinic receptor subunits hereby potentially interact with adaptor proteins to bind to discs large (Dlg). Importantly, we identify elevated α2 subunit dynamics in the context of associative (M4; Kenyon cells [KC] and dopaminergic neuron [DAN] activation needed for memory formation) and non-associative (α′3 MBONs; odor activates both KCs and DANs) memory expression. Increased α2 subunit dynamics in both cases are triggered by odor application. At the level of M4, suppressed dynamics, would correspond to postsynaptic depression, while at the level of α′3 MBONs increased dynamics could result in postsynaptic depression. Therefore, different learning rules might govern the incorporation, exchange, or stabilization of receptors in or out of synapses. Please see Discussion for further details. Bottom panel: Proposed time line of molecular correlates and experimental read-outs of memory induction and expression (please see Discussion for details).

MBONs do not appear to exhibit prominent spines on their dendrites (*Takemura et al., 2017*) (but see section below: Are cholinergic and glutamatergic synapses interchangeable?). Therefore, increased dendritic activation could lead to a change in membrane resistance and result in synaptic interference.

It should also be noted that M4, which shows depression (*Figure 5*), and M6 have common but also distinct physiological roles, for instance, during aversive memory extinction (*Felsenberg et al., 2018*). Besides that, different temporal requirements for M4 and M6 memory expression have been reported (*Bouzaiane et al., 2015*). It is therefore possible that physiological changes in the context of appetitive learning lead to different plasticity profiles in M4 and M6 neurons, respectively, or that initial potentiation over time can be reverted to depression. As noted above, MBON drive is bidirectionally modifiable and has the propensity to both potentiate and depress (*Owald et al., 2015*; *Bouzaiane et al., 2015*; *Lewis et al., 2015*; *Handler et al., 2019*). It remains unclear, whether the applied protocols would elicit plasticity (and if so depression or facilitation) at the M4 dendrites, which is difficult to assess with our experimental design. In summary, the observed ex vivo plasticity trace (*Figure 3*) should solely be viewed as a proof of principle that postsynaptic (MBON) plasticity can take place without presynaptic (KC) contribution per se.

Local acetylcholine application to the MB can also activate calcium transients in dopaminergic presynaptic terminals (*Cervantes-Sandoval et al., 2017*). Therefore, our protocol could in principle include some dopaminergic contributions already at the pre-training level. However, control experiments using the paired training protocol in the absence of CsChrimson expression in DANs, do not show any signs of plasticity (*Figure 3* and *Figure 3—figure supplement 1*). Moreover, it has been previously demonstrated that, to actually release dopamine from the presynaptic terminal, a coincident signal via carbon monoxide is required (*Ueno et al., 2017*; *Ueno et al., 2020*). Therefore, an unwanted activation of DANs in our experiments is unlikely.

Previous studies have shown that loss of DopR (dDA1) causes aversive and appetitive memory impairments (*Kim et al., 2007*; *Qin et al., 2012*). Intriguingly, specifically re-expressing DopR in KCs rescued loss of both types of memory (*Kim et al., 2007*). However, while the reported memory impairments in *dopR*-deficient animals were strong for aversive memories, they were only partial for appetitive memories, indicating that appetitive memory traces could be mediated via other dopamine receptors at the MBON level. Future experiments will need to investigate which dopamine receptors are required at the level of the MBONs as well as the in vivo time course of dopaminergic signaling.

## Nicotinic receptors could follow defined temporal sequences to mediate memory expression

Lasting plasticity traces as observed here (*Figure 3*) appear to fit the core criteria for long-term potentiation of vertebrate glutamatergic postsynapses (*Nicoll, 2017*; *Bliss and Lomo, 1973*). Plasticity can be divided into an 'induction' (*Nicoll, 2017*) period mediated via NMDARs and a subsequent 'expression' (*Nicoll, 2017*) period that requires altered AMPAR dynamics (*Nicoll, 2017*). Our findings here lead to a model, where the nicotinic α5 subunit is required for the induction of appetitive memories at *Drosophila* MBONs (*Figure 2* [affecting 'immediate' appetitive memory], *Figure 9*). We propose that α5 nAChR subunits (that can form homomeric channels *Lansdell et al., 2012*; *Eadaim et al., 2020*)

could take on a similar role to NMDARs. α5 would gate the potentiation or depression of synaptic strength influencing the incorporation or exchange of additional receptor subunits or complexes. In line with this, we show that knockdown of α5 subunits interferes with familiarity learning in the α'3 compartment of the MBs: flies no longer form familiarity memories, they react to a novel odor the same way as to a familiar one, 'as if they had learned that this new odor was familiar before' (**Figures 7 and 8**). Moreover, we do not observe α5 subunit dynamics (**Figure 7**), whereas knockdown of α5 leads to decreased levels of α2 subunits (**Figure 4**), and α2 dynamics are no longer observable when knocking down α5 in the MBONs of the α'3 compartment (**Figure 7**). Thus, we can draw first analogies to glutamatergic systems governing plasticity in vertebrates. Whether more core criteria are met for the comparison of invertebrate and vertebrate plasticity systems, further depends on whether the here observed receptor dynamics will actually translate to exo-/endocytosis of postsynaptic receptors or lateral diffusion of receptor subunits along the MBON dendrites (see **Figure 9**). Our established system should provide the means to investigate this further in the future.

We also find that later forms of appetitive memory expression require both the α2 and α1 receptor subunits (**Figure 2**). A recent study (**Ihara et al., 2020**) has demonstrated that, when expressed heterologously, these subunits can co-assemble to form heterodimers with β subunits, which, depending on the precise composition of these channels, can harbor different properties, potentially reminiscent to AMPAR (**Greger et al., 2017**). However, MB distribution profiles of α1 and α2 subunits do not match completely, for instance, at the level of the γ5 or α'2 compartments (**Figure 4**), indicating that they could also partake in different or independent receptor configurations.

In order to dissect distinct roles for receptor plasticity in memory induction and expression, we experimentally probed several time points during associative appetitive memory formation. First, we probed 30-min memory following KC block to invariantly interfere with the memory acquisition and not the retrieval stage (**Figure 1**). Second, we probed immediate and 3-hr memory performance following receptor knockdown to distinguish between memory induction and memory expression requirements (**Figure 2**). Third, we investigate the time course of receptor dynamics during memory expression following memory induction with a resolution of 5 to 10-min intervals after artificial training (**Figures 6 and 7**). The overarching picture indicates that, indeed, directly following training, memory induction requires α5. Subsequently, at the resolution of minutes, regulation of α2 levels contributes to memory expression. While we cannot resolve the temporal time course at the level of T-maze behavior (**Figure 2**) or FRAP experiments (**Figure 6**) below several minutes, our in vivo training data (**Figure 5**) suggests that α2 requirement already becomes apparent within 1 min after training.

We show that familiarity learning, a specific form of habituation encoded at a higher-order integration center, the MBs, can take place when knocking down α2 nAChR subunits in α'3 MBONs in principle (**Figure 8**), however, at clearly decreased efficacy and only after several trials. We speculate that the observation of memories still being expressed per se in this context could be explained by redundancies with α1 or other subunits (but see heterogeneous localization and enrichment in different MB compartments, **Figure 4**). Redundancies could also explain why we partially observe functional phenotypes after knockdown of individual subunits but only moderate structural changes. We would also like to point out that subunits we did not identify absolutely as required for memory expression (**Figure 2**) in this study could nonetheless partake in distinct phases of plasticity processes.

In the context of both familiarity learning and appetitive conditioning, odor exposure induces increased α2 subunit dynamics (**Figures 6 and 7**) accompanying postsynaptic depression (**Owald et al., 2015**; **Hattori et al., 2017**; **Figure 7**), while not or mildly affecting α5 subunits (for familiarity learning). Therefore, the same basic mechanisms, odor-induced α2 receptor dynamics, seem to express two opposed plastic outcomes in the context of associative and non-associative memories and contribute to different learning rules across MB compartments (**Hige et al., 2015**; **Aso and Rubin, 2016**). We speculate that α2 dynamics induced by odor in the M4/6 dendrites could be reminiscent of dark currents in the vertebrate visual system (**Hagins et al., 1970**), allowing for rapid adaptation with low levels of synaptic noise. Receptor exchange at the level of M4/6 dendrites would actually take place when no associations are formed and stalled when DANs (triggered by sugar) are simultaneously active with KCs (triggered by odor). Indeed, repeated OCT stimulation (**Figure 5—figure supplement 1**) led to a facilitation of calcium transients (potentially corresponding to an increase of receptor incorporation), while depression (in this case likely to be mediated by removal of receptors, but see above) is triggered by paired training (**Figure 5**). In contrast, at the level of the α'3 compartments, odor

activates both MBONs and DANs. Here, the plasticity rule would be reversed. Synaptic depression is accompanied by actively changing the receptor composite. We speculate that such plasticity could function reminiscent of mechanisms observed for climbing fiber-induced depression of parallel fiber to Purkinje cell synapses (*Ito, 2001*). However, whether increased dynamics can be translated to more incorporation or removal of α2-type receptors, or depending on the plasticity rule both, will require high-resolution imaging experiments in the future.

## Are cholinergic and glutamatergic synapses interchangeable?

Our study fuels the question of how unique properties of individual neurotransmitter systems at synapses are. While dopamine signaling is remarkably conserved between invertebrates and verte-brates, cholinergic and glutamatergic systems appear, now more than before (with this study), some-what interchangeable. While vertebrates (but also evolutionarily distant *Caenorhabditis elegans*), for instance, use acetylcholine at the neuromuscular junction and store memories predominantly at glutamatergic synapses, it is the other way around in *Drosophila* and other invertebrates, such as *Sepia* (*Barnstedt et al., 2016*; *Owald and Waddell, 2015*; *Shomrat et al., 2011*; *Ackermann et al., 2015*; *Owald and Sigrist, 2009*). Now we show that, at cholinergic synapses, α5 and α2 subunits, at least to a certain extent, behave in a potentially comparable way to NMDARs and AMPARs at gluta-matergic synapses during postsynaptic plasticity which underlies memory storage. In this context, we offer several lines of evidence that invertebrates utilize postsynaptic plasticity during memory storage.

We therefore propose that, across phyla, postsynaptic plasticity, with the propensity to store memo-ries and adapt network function plastically, can take place regardless of neurotransmitter identity.

One key difference between the dendritic arbors of the MBONs analyzed (e.g. *Figure 10a*) in this study compared to dendrites of glutamatergic neurons in vertebrates, is a lack of anatomical spines (*Figure 10b*). Without spines, how can input specificity be preserved at MBON postsynaptic densi-ties? KC output to MBON input analysis of the recently published fly hemibrain connectome (*Scheffer et al., 2020*; neuprint.org; *Clements et al., 2020*) suggests that, at the ultrastructural level, MBON postsynaptic densities are separated spatially (*Figure 10b*). Compartmentalization could therefore be mediated by, for instance, biochemical separation of PSDs. Importantly, input-specific plasticity has been shown to be inducible in non-spiny neurons in vertebrates, with diffusion barriers established, e.g., through calcium buffers, between postsynaptic densities (*Goldberg et al., 2003*).

Interestingly, our MBON input analysis further revealed that postsynaptic plasticity mechanisms could actually add a layer to promote input specificity. Indeed, we find that single presynaptic KC release sites that innervate MBON dendrites can also target other MBONs and/or other postsynaptic targets simultaneously (*Figure 10c–g*). Plasticity confined to single postsynaptic densities innervated by a KC terminal could therefore change the weight of transmission for one target (e.g. MBONs involved in memory storage, such as M4 *Owald et al., 2015*), while not changing the weight of the connection to other targets (e.g. MBONs not involved in a specific action or other targets of non-MBON identity, *Figure 10h*). It should be noted that this architecture does not exclude presynaptic plasticity mechanisms (*Stahl et al., 2022*; for instance, following aversive conditioning). Indeed, we would speculate that synaptic connections can be subdivided into distinct compartments on both the pre- and the postsynaptic side, potentially through transsynaptic molecules (*Owald and Sigrist, 2009*; *Tang et al., 2016*), allowing for fine-tuned and target-dependent changes of parameters within either side of a synapse.

Together, we propose a model (*Figure 9*) in which α5 subunit containing receptors could mediate the early phase of postsynaptic memory storage, and we speculate this could lead to elevated post-synaptic calcium flux (not addressed in this study). Concurrent events see changed dynamics of the α2 receptor. Nicotinic receptor subunits hereby could interact with adaptor proteins to bind to Dlg, remi-niscent to what is known for AMPAR (*Won et al., 2017*). Importantly, we identify elevated α2 subunit dynamics in the context of associative and non-associative memory expression. Increased α2 subunit dynamics in both cases are triggered by odor application. At the level of M4/6, suppressed dynamics would correspond to synaptic depression, while at the level of α'3 MBONs increased dynamics may result in postsynaptic depression. Therefore, different learning rules could govern the incorporation or exchange or mobilization of receptors in or out of synapses. The precise molecular and biophysical parameters underlying these plasticity rules are currently unknown and will need to be addressed in

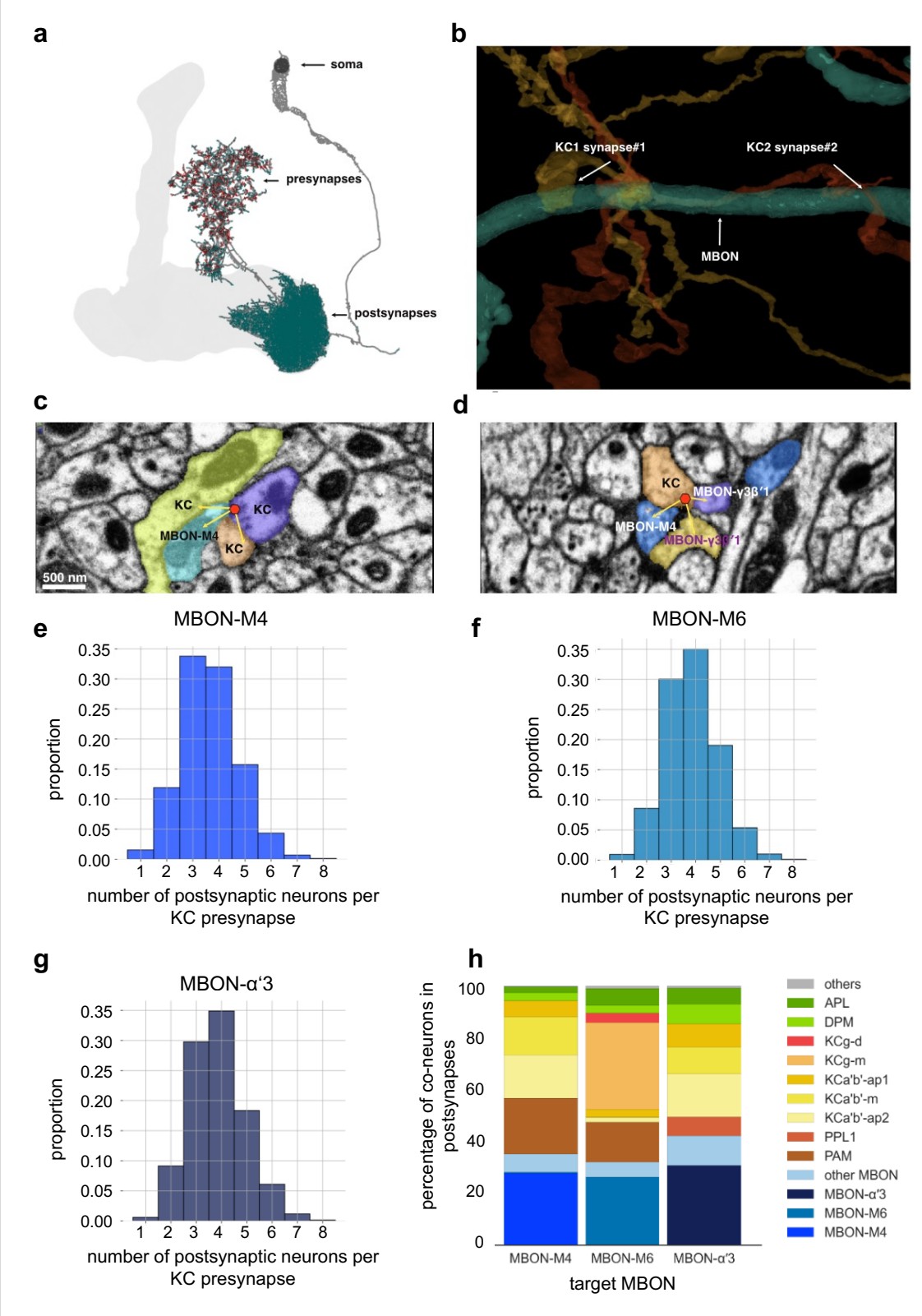

**Figure 10.** Mushroom body (MB) output connectomics (**a**) Example MB output neuron (MBON; here: M6) with pre- and postsynapses labeled. (**b**) Reconstructed example synapses from electron microscopic (EM) volume (neuprint.org): two different Kenyon cells (KCs) connect to the same MBON on the postsynaptic side. (**c**) EM image (neuprint.org) showing a KC presynapse simultaneously innervating two other KCs and the M4 MBON. Scale bar: 500 nm. (**d**) EM image (neuprint.org) showing a KC presynapse simultaneously innervating the M4 MBON and two sites of another MBON. (**e**) Analysis of

*Figure 10 continued on next page*

*Figure 10 continued*

number of postsynaptic partners for each KC presynapse identified providing input to M4. The histogram shows the distribution of KC synapses to M4 relative to how many postsynaptic partners the KC presynapses have. (**f**) Analysis of number of postsynaptic partners for each KC presynapse identified providing input to M6. The histogram shows the distribution of KC synapses to M6 relative to how many postsynaptic partners the KC presynapses have. (**g**) Analysis of number of postsynaptic partners for each KC presynapse identified providing input to α'3 MBONs (pooled). The histogram shows the distribution of KC synapses to α'3 MBONs relative to how many postsynaptic partners the KC presynapses have. (**h**) Percentage of types of neurons that share a KC presynapse with a given MBON.

the future. One option could include potential exchange of α2 subunits for a receptor complex with higher calcium permeability.

Our findings are consistent with the current MB skew model (*Owald and Waddell, 2015*), where the summed MBON output will determine an animal's choice. However, we add an additional layer, already at the MBON input site. Changes do not happen at the presynaptic compartment only but potentially at both synaptic compartments. Thus, the power to store (potentially conflicting) information separately at either the pre- or postsynaptic site, equips the system with additional flexibility. How precisely pre- to postsynaptic and post- to presynaptic signaling is regulated will need to be addressed in the future but will likely involve transsynaptic signaling routes (*Owald and Sigrist, 2009*; *Tang et al., 2016*). Importantly, the identified modes of postsynaptic plasticity will open avenues for investigations looking into pre- versus postsynaptic contributions during reversal learning, reconsolidation, and extinction learning (*Felsenberg et al., 2017*; *Lüscher and Malenka, 2011*).

## Methods
### Fly genetics

Flies were raised on standard food under standard laboratory conditions unless stated otherwise (25°C, 65%, 12-hr light-dark cycle; *Owald et al., 2015*; *Raccuglia et al., 2019*). Driver lines used were MB011B (Split-Gal4) (*Aso et al., 2014a*), MB112C (*Aso et al., 2014a*) (Split-Gal4), MB461B (*Aso et al., 2014a*) (Split-Gal4), MB027B (*Aso et al., 2014a*) (Split-Gal4), R13F02-Gal4 (*Aso et al., 2014a*), OK107-Gal4 (*Barnstedt et al., 2016*), VT1211-GAL4 (*Owald et al., 2015*), and R58E02-LexA (*Handler et al., 2019*). We used the following UAS-nAChR[RNAi] flies (*Barnstedt et al., 2016*; *Cervantes-Sandoval et al., 2017*): Bloomington stock numbers 28688, 27493, 27671, 31985, 25943, 27251, and 25835. Additionally, we used (*Bachmann et al., 2004*; *Soukup et al., 2013*) Dlg[S97]-RNAi as well as UAS-Dlg[GFP], tubP-GAL80[ts] (*Perisse et al., 2013*), UAS-Shi[ts1] (*Perisse et al., 2013*), 247-dsRed (*Owald et al., 2015*), LexAop-CsChrimson, UAS-GCaMP6f (*Barnstedt et al., 2016*), and UAS-SynaptoPhluorin (*Lin et al., 2014a*). Note that complex genotypes did not always permit usage of MB011B for genetical access to M4/6 neurons throughout the manuscript. In that case, in order to reduce genetic complexity, we used VT1211-Gal4.

### Behavior
#### T-maze memory

3–9 days old mixed-sex populations were trained and tested together as previously described (*Owald et al., 2015*). Odors used were 3-octanol (OCT, Aldrich) and 4-methylcyclohexanol (MCH, Aldrich) diluted in mineral oil (approximately 1:100 for aversive, 1:1000 for appetitive memory, absolute concentrations were minimally adjusted to prevent odor bias). For aversive protocols, flies were exposed to the CS+ for 1 min with 12 1.5 s long 120 V electric shocks (interstimulus interval: 3.5 s) followed by 45 s of air, 1 min of CS− exposure and another 30 s of air. Flies were given 2 min to choose between the CS+ and CS− in a T-Maze during retrieval in the dark. For appetitive conditioning, flies were starved for 20–24 hr before the experiment. Flies were exposed to the CS− for 2 min. After 30 s, flies were exposed to the CS+ paired with sugar for 2 min followed by another 30 s of air. Performance indices were calculated as described previously (*Owald et al., 2015*). Time of retrieval is stated in the figures. For Shi[ts] experiments, flies were kept at 32°C 30 min prior to and during training and brought to room temperature directly afterward. Room temperature was approximately 23°C. For *Figure 2* and *Figure 2—figure supplement 1*, behavioral data sets from separate experiments were pooled. Note that 'screening hit' data displayed in *Figure 2a and b* and *Figure 2—figure supplement 1a, b* were replotted to allow for comparison of genotypes with the corresponding genetic controls in

*Figure 2—figure supplement 2e-i*. Gal80$^{ts}$ flies were always raised at 18–20°C and were placed at 32°C 3–5 days before the experiment (*Figure 2*) or were kept at 18–20°C throughout (*Figure 2— figure supplement 2*).

## Familiarity learning

Familiarity training was essentially performed as described before (*Hattori et al., 2017*) with slight adjustments. Flies were covered in yellow dust (*Hattori et al., 2017*) (Reactive Yellow 86, Fisher Scientific) and placed in a cylindrical custom designed chamber. To ensure a constant air stream, we placed the chamber between an air and a vacuum pump (800 ml/min). Air permeable cotton wool was used to close the open ends of the chamber. The air supply was either connected to pure mineral oil or MCH diluted in mineral oil at a concentration of 1:50. For switching between odor and mineral oil, a clamp was manually opened and closed. Video recording was performed at 26 frames per second. For recordings and analyses we used Python (v3.6) in Anaconda Jupyter Notebook environment.

## Imaging

### Confocal single-photon imaging and receptor quantification

#### Fixed explant brain imaging

Brains were dissected on ice, fixed in 4% paraformaldehyde (Sigma) for 20 min, and placed in PBST (phosphate-buffered saline, 0.1% Triton) for 30 min followed by washing with PBS for 20 min twice. Vectashield was used as mounting medium. Flies were 2–8-day-old females raised at room temperature.

#### Recording endogenous fluorescence

Imaging was performed using a confocal single-photon inverse microscope (Leica SP5/STED) equipped with a ×64 oil objective. Laser power and gain were adjusted between experiments, making normalization of the signals necessary. Values for the heatmap in *Figure 4* were normalized to the mean MB fluorescence value to ensure comparability. Voxel size was (height × width × depth) 123 nm × 123 nm × 500 nm. ROIs were drawn manually in ImageJ using the 247-dsRed channel for orientation (*Figure 4a*). Heat maps were created in Microsoft Excel. For quantifications following knockdown, the γ5 compartment was normalized to γ4 ((γ5 − γ4)/ γ4), and the β'2 to the β'1 compartment ((β'2 − β'1)/ β'2) of the same animal. Each 'n' corresponds to one hemisphere.

### In vivo two-photon calcium imaging

To measure odor responses, female 3–6-day old flies expressing UAS-GCaMP6f alone or together with UAS-RNAi to α2 or α5 at the level of M4/6 were tethered under the multiphoton microscope (Femtonics), essentially as described before (*Owald et al., 2015*; *Böhme et al., 2019*). To measure M4/6 responses to odor presentation alone, five alternating 1 s OCT and MCH puffs were applied with 30 s in between each presentation. Fluorescent signals were recorded from dendrites in the β'2 MB compartment using MESc software (Femtonics) at a frame rate of roughly 30 Hz. ROIs incorporating the dendritic arbors were manually drawn. Data was processed using a Savitzky-Golay filter. Further analysis was performed using Matlab. For absolute training, we imaged from axonal arbors, and the following protocol was applied: after initial testing for odor responses, starved flies were exposed to 1-s odor puffs (MCH) twice with a 30 s gap between the applications. Corresponding odor responses were averaged. Paired training consisted of odor application (five consecutive 1-s odor puffs [MCH] with 6 s gap), while the fly fed on a sucrose droplet (saturated solution) for 30 s delivered by a custom-made feeding arm (*Lin et al., 2014b*). Unpaired training consisted of 30 s sucrose feeding followed by 30 s of odor applications as described above but with a delay of 45 s. The training was monitored with a video camera to verify accurate feeding of the fly. After a 1-min break, two odor puffs with a gap of 30 s were applied. Again, corresponding odor responses were averaged. Peak responses and areas under the curve (AUCs) were calculated using R and the first 3 s following odor onset were analyzed in order to cover entire responses. The AUCs pre- and post-training were normalized to the mean pre-training values of a group, respectively. The averaged test responses pre-training were compared to the average post-training responses (9–12 hemispheres were analyzed from 5 to 7 animals per genotype and condition) using a paired t-test or Wilcoxon matched-pairs signed rank test.

## In vivo confocal single-photon imaging of receptor dynamics and calcium transients

3–4 days after enclosure, female flies were prepared as described above and imaged. Imaging was performed using a SP5 single-photon confocal microscope (Leica microsystems). Recording frame rate was 3 Hz. For bleaching high laser power was focused on the α'3 compartments for 15–25 s. The baseline was recorded after bleaching, immediately before fixed inter-stimulus interval training (*Hattori et al., 2017*). OCT was presented 10 times for a second with a 6-s pause in between. Odor delivery (CON electronics) was controlled by the Leica acquisition software. After training, the same brain plane was recorded for 10 s with a pixel size of 200 nm in time intervals of 0, 5, 10, 15, 20, 30, and 60 min after training. For control experiments, air only was delivered to the chamber. Images of the same time interval recordings were averaged and processed in ImageJ. Gaussian blur ($\sigma$=0.5) was applied for smoothing, and ROIs were selected manually.

## In vivo two-photon imaging of receptor dynamics

FRAP experiments were performed in vivo. 2–8-day-old flies were anesthetized on ice and mounted in a custom-made chamber. The head capsule was opened under room temperature sugar-free HL3-like saline, and legs were immobilized with wax. Sugar-free HL3-like saline containing 30 units of papain (Roche) was applied to the head capsule for 8 min to digest the brain's glial sheath and facilitate removal. Images were acquired using a multiphoton microscope (Nikon) with a ×25 water-immersion objective, controlled by Nikon NIS Elements software. Diluted odors (MCH or OCT in mineral oil 1:1000) were delivered on a clean air carrier stream using a six-channel delivery system (CON electronics). The flies were subjected to experimental conditions including either no odor (air), odor only, odor paired with local dopamine (10 mM) injection via a micropipette, or local dopamine injection only (see *Figure 6b* and *Figure 6—figure supplement 1b* for experimental protocol schematics). Photobleaching was accomplished using focused, high intensity laser exposure for ~1 min. Analysis of fluorescence recovery was performed using FIJI. ROIs were manually selected, and the percent recovery fluorescence was calculated by subtraction of the post-bleaching baseline fluorescence and division by the pre-bleaching baseline fluorescence. To fit the inverse exponential decay that is expected for FRAP data, we first inverted the percent fluorescence recovery values by subtracting them from 1 and then log-transformed the resulting values. These log-transformed values were used in a linear mixed effects model without intercept using the interaction between condition and time as fixed effect - to determine condition-specific differences of the recovery kinetics - and time as random effect (R package lme4). A linear mixed effect model was used to appropriately model repeated measures within animals. By inverting and log-transforming the fluorescence recovery values, this approach is equivalent to fitting an inverted exponential decay function. For plotting, all values including regression coefficients were back-transformed to the original scale. Significance of recovery of individual conditions was assessed using the regression coefficients of the condition-time interaction of the linear mixed model. Differences of recovery between pairs of conditions were tested using pairwise comparisons of estimated marginal means of the linear mixed model (R package emmeans). Correction for multiple pairwise comparisons was performed using Tukey's method.

## Explant brain widefield imaging, neurotransmitter application, and optogenetics

### Postsynaptic plasticity induction

Brains of 3–10-day-old mixed sex flies were dissected on ice. Flies expressed CsChrimson[tdTomato] under control of R58E02-LexA and UAS-GCaMP6f (and α2 RNAi) under control of MB011B. The head capsule and sheath were removed in carbogenated solution (103 mM NaCl, 3 mM KCl, 5 mM N-Tris, 10 mM trehalose, 10 mM glucose, 7 mM sucrose, 26 mM NaHCO$_3$, 1 mM NaH$_2$PO$_4$, 1.5 mM CaCl$_2$, 4 mM MgCl$_2$, 295 mOsm, pH 7.3) with forceps. The brain was subsequently perfused with carbogenated solution containing TTX (2 μM; 20 ml/10 min flow speed) and imaged using an Olympus MX51WI wide field microscope with a ×40 Olympus LUMPLFLN objective and an Andor iXON Ultra camera controlled by Solis software. An Olympus U25ND25 light filter was placed in the beam path to minimize baseline CsChrimson activation. A custom designed glass microcapillary was loaded with uncarbongenated solution containing 0.1 mM acetylcholine and maneuvered to the M6 dendrites.

The injection pressure of a P25-1-900 picospritzer was calibrated between 3 and 8 psi (depending on initial calcium responses). Each local acetylcholine application spanned 15 ms with a 4 s inter-injection interval.

Three pulses of acetylcholine followed by a 2–3-min break were recorded after which the optogenetic response was assessed by applying 2 s red light pulses with an inter-red light-interval of 4 s. Three acetylcholine pulses were recorded followed by either five acetylcholine injections, five red light pulses, or both paired. For paired training, both stimuli began simultaneously and the acetylcholine injection lasted for 15 ms (and gave rise to a calcium transients typically lasting >1 s, please see example in *Figure 3*), while the paired red light pulse lasts for 2 s, allowing for maximal temporal overlap. This process was repeated 5×, with a 4-s break between trials. Following the training trial, three final acetylcholine test injections were applied after 1 min. For analysis, the first of the three acetylcholine injections was always discarded because of initial dilution of the capillary tip, and the remaining two peak intensities were averaged.

All peaks within an experiment were quantified relative to the fluorescence baseline that we calculated for pre- and post-training acetylcholine responses. Baselines were set independently for each pre- and post-training recording using the polynomial interpolation function in NOSA (neuro-optical signal analysis) (*Oltmanns et al., 2020*). For investigating α2 knockdown, only the paired condition was tested.

For controls not expressing CsChrimson, we used VT1211-Gal4 driving UAS-GCaMP6f, instead of MB011B, for technical reasons. This was combined with either expression of R58E02-LexA or LexAop-CsChrimson^tdTomato. Only paired training was investigated in this context.

## Excitability of KC axons

To test whether KC axons were excited by focal acetylcholine injections at the level of M4/6 dendrite innervation, either UAS-SynaptoPhluorin or UAS-GCaMP6f was expressed under the control of OK107-Gal4. Following the acetylcholine injection experiment, the capillary was exchanged with a capillary containing the same solution with additional 300–400 mM KCl, to evaluate tissue health (not shown for GCaMP6f imaging). To pick up potentially small changes, we increased the injection pressure to 8–14 Psi and the injection time to 150–225 ms (GCaMP6f, 8 s inter-injection interval and three consecutive injections) and 300–525 ms (SynaptoPhluorin, with an 8 s inter-injection interval and three consecutive injections).

Images were analyzed using NOSA (*Raccuglia et al., 2019*; *Oltmanns et al., 2020*) and GraphPad Prism.

## Statistics

Statistical analyses were performed as stated in the previous method sections and figure legends. Data were always tested for normality using a Shapiro-Wilk test. If normally distributed, data were analyzed using ANOVA followed by post-hoc test or a (paired) t-test. If not normal, we used a Kruskal Wallis followed by post-hoc test, or a Wilcoxon matched-pairs signed rank test.

## Tagged receptor subunits

All subunits were tagged using CRISPR technology and motifs previously described (*Raghu et al., 2009*). All EGFP-tagged AChR α subnunits were generated by WellGenetics Inc (Taiwan) using

**Table 1.** Used EGFP positions and gRNA sequences of this study.

| Subunit | EGFP position between AA | gRNA sequence |
| --- | --- | --- |
| nAChRalpha1/CG5610 | 438D and 439L | ACAGATCGTCGTCGGCGCCC[GGG] |
| nAChRalpha2/CG6844 | 456G and 457L | CAGATTCAGCGGCTTGGTGG[GGG] |
| nAChRalpha4/CG12414 | 426M and 427D | AATAGCCGCCGTCCCCGATA[TGG] |
| AChRalpha5/CG32975 | 717G and 718S | CAGCACCCGAATGCCGGATG[CGG] |
| nAChRalpha6/CG4128 | 403T and 404A | TTACGCCGACGAGCCAATGG[CGG] |
| nAChRalpha7/CG32538 | 464G and 465S | GCAAGGGGATGACGGCAGCG[TGG] |

CRISPR-based mutagenesis akin to Kondo and Ueda (*Kondo and Ueda, 2013*). Please see *Table 1* for position and sequence.

In brief, nAChR gRNA sequence was cloned into an U6 promoter plasmid. Cassette EGFP-PBacDsRed containing EGFP and 3xP3-DsRed and two nAChR α-homology arms were cloned into pUC57-Kan as donor template for repair.

nAChR alpha-targeting gRNAs and hs-Cas9 were supplied in DNA plasmids together with donor plasmid for microinjection into embryos of the w1118 strain. F1 flies carrying the 3xP3-DsRed selection marker were further validated by genomic PCR and sequencing. The 3XP3-DsRed marker was excised by Piggy Bac transposase leaving an exogenous 2aa linker of Valine and Lysine (GTTAAA) after excision.

## Connectome analysis

In *Figure 10*, we analyzed the partial connectome of the female adult fly brain (hemibrain v1.2.1; *Scheffer et al., 2020*), using the neuprint-python package (https://github.com/connectome-neuprint/neuprint-python; *Berg et al., 2022*). To investigate the synaptic relationship between KC presynapses and MBON postsynaptic sites, we first identified KCs with the status 'Traced' in the connectome.

Second, for each MBON of interest, we identified the relevant KCs connected to it (which is a subset of the count in the previous step). Third, for each KC identified, we selected each presynaptic terminal (x, y, and z locations of synapses) connected to the MBON of interest, and for each of these presynapses, we identified all synaptic partners residing on the postsynaptic side. Fourth, for the MBON of interest, we counted the number of postsynaptic connections per individual presynapses (that also contain the specific MBON, *Figure 10e–g*). Finally, we identified the composition of all neurons identified as co-postsynaptic partners of KC to MBON synapses.

## Acknowledgements

We thank Anatoli Ender, Johannes Felsenberg, Davide Raccuglia, Stephan Sigrist, Uli Thomas, and Scott Waddell for comments on the manuscript, Stephan Sigrist and Uli Thomas for reagents, the Janelia and Vienna fly projects, and the Bloomington Stock Center and VDRC for fly lines as well as Daisuke Hattori and Yoshi Aso for help with the familiarity experiments and Julia Thüringer for help with analysis. Multiphoton and single-photon confocal imaging were partially performed using microscopes of the AMBIO and NWFZ core facilities of the Charité. We would like to acknowledge scidraw. io for following drawings: flies (http://org/10.5281/zenodo.3926137, https://doi.org/10.5281/zenodo.3925939), brain (http://10.5281/zenodo.4420079), and objective (http://10.5281/zenodo.4914800). Funding: Funded by the Deutsche Forschungsgemeinschaft (DFG, German Research Foundation) under Germany's Excellence Strategy – EXC-2049–390688087, the Emmy Noether Programme, TP A27 of SFB958 (184695641), TP A07 of SFB1315 (327654276) as well as TP A05 of FOR (365082554) to DO. SRJ is supported by the Walter Benjamin Programme of the DFG.

## Additional information

### Funding

| Funder | Grant reference number | Author |
| --- | --- | --- |
| Deutsche Forschungsgemeinschaft | EXC-2049-390688087 | David Owald |
| Deutsche Forschungsgemeinschaft | 184695641 | David Owald |
| Deutsche Forschungsgemeinschaft | 327654276 | David Owald |
| Deutsche Forschungsgemeinschaft | 365082554 | David Owald |
| Deutsche Forschungsgemeinschaft | 467545627 | Sridhar R Jagannathan |

| Funder | Grant reference number | Author |
|---|---|---|
| Deutsche Forschungsgemeinschaft | Emmy Noether Programme | David Owald |

The funders had no role in study design, data collection and interpretation, or the decision to submit the work for publication.

## Author contributions

Carlotta Pribbenow, Conceptualization, Data curation, Formal analysis, Investigation, Writing – original draft, Writing – review and editing; Yi-chun Chen, Isabella Balles, Tania Fernández-d V Alquicira, Carolin Rauch, Sridhar R Jagannathan, Data curation, Formal analysis, Investigation, Writing – review and editing; M-Marcel Heim, Eric Reynolds, Data curation, Formal analysis, Investigation, Visualization, Writing – review and editing; Desiree Laber, Data curation, Investigation, Writing – review and editing; Silas Reubold, Data curation, Investigation, Visualization, Methodology, Writing – review and editing; Raquel Suárez-Grimalt, Jörg Rösner, Investigation, Methodology, Writing – review and editing; Lisa Scheunemann, Investigation, Writing – review and editing; Tanja Matkovic, Investigation; Gregor Lichtner, Formal analysis, Investigation, Visualization, Writing – review and editing; David Owald, Conceptualization, Resources, Supervision, Funding acquisition, Validation, Investigation, Visualization, Writing – original draft, Project administration, Writing – review and editing

## Author ORCIDs

Carlotta Pribbenow ⓘ http://orcid.org/0000-0002-1893-6435
Yi-chun Chen ⓘ http://orcid.org/0000-0002-9187-930X
M-Marcel Heim ⓘ http://orcid.org/0000-0002-8079-4204
Silas Reubold ⓘ http://orcid.org/0000-0002-3308-3733
Eric Reynolds ⓘ http://orcid.org/0000-0001-8597-6173
Tania Fernández-d V Alquicira ⓘ http://orcid.org/0000-0003-2462-1664
Raquel Suárez-Grimalt ⓘ http://orcid.org/0000-0002-5374-7963
Lisa Scheunemann ⓘ http://orcid.org/0000-0002-6013-388X
Gregor Lichtner ⓘ http://orcid.org/0000-0002-5890-1958
Sridhar R Jagannathan ⓘ http://orcid.org/0000-0002-2078-1145
David Owald ⓘ http://orcid.org/0000-0001-7747-7884

## Decision letter and Author response

Decision letter https://doi.org/10.7554/eLife.80445.sa1
Author response https://doi.org/10.7554/eLife.80445.sa2

# Additional files

## Supplementary files
- MDAR checklist
- Supplementary file 1. Supplementary statistics.
- Source data 1. Source data.

## Data availability

All data generated or analysed during this study are included in the manuscript and supporting files. The code used in this study is available on GitHub, (copy archived at swh:1:rev:0550ee981ec5b6e-b7e76e9062e3c9c8dead42306; *Jagannathan, 2022*). Materials can be requested from the corresponding author.

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
