## [Editor Report]

Learning-dependent plasticity is thought to take place predominantly presynaptically in Drosophila. This paper by the Owald group adds important evidence for postsynaptic plasticity mechanisms, including that appetitive memory is impaired when nAChR subunits (α2, α5) and scaffold protein Dlg are downregulated in specific mushroom body output neurons. In a tour-de-force, they combine physiology, *Drosophila* genetics, and behaviour and the work also emphasises the similarities in learning and memory mechanisms between vertebrates and invertebrates.

---

## [Decision Letter]

**Decision letter after peer review:**

Thank you for submitting your article "Postsynaptic plasticity of cholinergic synapses underlies the induction and expression of appetitive memories in *Drosophila*" for consideration by *eLife*. Your article has been reviewed by 3 peer reviewers, and the evaluation has been overseen by a Reviewing Editor and K VijayRaghavan as the Senior Editor. The following individual involved in the review of your submission has agreed to reveal their identity: Simon G Sprecher (Reviewer #1).

Essential revisions:

1. Please revise the manuscript focusing on and removing non-essential parts to improve readability (eg add explanations where needed for non-experts, fly genotypes,…). Also, provide a better scholarly referenced background into past work and literature. It felt sometimes that too much is trying to be said, without sufficient depth.

2. An important point that we would want you to address is the second point of Reviewer 3's list of comments where they ask to compare the difference between paired and unpaired odour responses during the test. Please also consider points 3, 5, and 7 of the same reviewer, but they may be addressed textually.

3. We ask you to include the necessary details regarding the CRISPR lines you included as "unpublished". This paper will effectively publish them.

4. All other points, including the clarifications asked for by reviewer 2 can be addressed textually.

*Reviewer #1 (Recommendations for the authors):*

Overall the manuscript describes an important advance, both experimentally as well as conceptually. However, there are a few points that I think should be taken into account:

– To follow the experiments perform I feel it is important to describe in the Results section the genotypes used in the experiments performed. This can also be done in brackets to not disrupt the flow of the text. It is clear that some experiments may use complicated genotypes, however, I feel it's important that at least the logic of the experiments is apparent from the text (which Gal4, with UAS, which LexA or temp/opto line in which condition temp/light).

– An important aspect of the manuscript relates to the differential use, localization, or expression of nAChR subunits. While the knock-down experiments described in figure 1 are easy to follow there is no information on how the expression was assessed, apart from the note that CRISPR was used as unpublished data. The authors should decide if they either want to include this set of experiments and then publish these lines or if they prefer not to publish these lines, but should then exclude this set of experiments. Without information about the locus, what has been modified, and what impact this may have it is not possible to assess these experiments properly. I am admittedly a bit dissatisfied by this section, particularly because of its central position in the logic.

– The explant optogenetic/thermogenetic/pharmacology experiments are intriguing and – to be frank- quite refreshing. While the authors rightfully point out some limitations am sure the authors are aware of the points that could have been addressed. I do not feel that it is necessary to perform additional experiments here (though of course I'd be curious about the results), however, I would encourage the authors to at least highlight some of the obvious steps that will quite certainly allow probing their model (e.g receptors, dopamine, etc).

*Reviewer #2 (Recommendations for the authors):*

As somebody not in the learning and memory field, nor with expertise in the structure and function of the insect mushroom body, I cannot provide a very helpful critique of their specific experiments. However, there are several points, from an outside point of view, which may help improve the presentation. In no particular order:

1) I think the authors should be more precise and explanatory in the terms they use. For example, what is the difference between relative familiarity with odors (non-associative familiarity learning) and habituation? Similarly, it would help if the paradigms for immediate, intermediate/3 hours, induction, and expression of memories are more clearly explained (perhaps even through schematics in one of the figures. This becomes important when they try to discuss these different forms with activities of different receptor subunits and at different MB compartments.

2) A schematic of the MB would definitely help, with a focus on the cellular aspects and the different compartments. Their broad drawings of KC and MBONs do not help, since the work emphasizes the sub-cellular nature of receptor dynamics in specific compartments with functional specializations. Similarly, in Supplementary Figure 9, they provide EM data, but this would be helped by a cellular level that shows soma, axons/dendritic projections.

3) How is synaptic weight defined? And how is their plasticity defined, since they use calcium transients as opposed to actual structural changes in the synapse? Many use GCamp to monitor calcium transient changes – do all these then imply "plasticity"?

4) There are many cases where different terms are used, and it is not clear to what extent these are similar words for the same process or different processes. For example, in line 272 there is this: "…postsynaptic sensitivity can change independently of the presynapse. Indeed, the observed postsynaptic potentiation was not observed…, consistent with postsynaptic plasticity…". Thus the word postsynaptic is followed by three different terms (sensitivity, potentiation, and plasticity). This is difficult for non-experts to understand. As a further example, in line 342, it would help if they explained "pre-potentiated synaptic transmission" versus "baseline transmission".

5) The Discussion part is divided into seven subheadings---this seems a bit lot in terms of what the major discussion points are. Seven major take-home messages seem like a lot to emphasize for a single paper. It may help to make this part a bit more concise.

6) The supplement to Figure 9 (EM data) has nothing to do with Figure 9 itself (a speculative model). Maybe the EM data can be put as part of regular Figure 9.

All in all, I think the paper is very difficult to understand for non-experts because there are so many different terms used and it's not clear what the precise differences are, whether these are fundamentally different concepts or just different terms for similar things. Also, there are numerous very precise statements and not enough references at appropriate places to back them up. For example, line 504: "It also argues against the assumption that appetitive and aversive memories will necessarily use the same molecular machinery to store information. Findings in the past, predominantly based on investigating aversive memories have been generalized to learning per se". On which/whose works are such assumptions based?

*Reviewer #3 (Recommendations for the authors):*

While the results are certainly interesting, I have a number of points for the authors to clarify before publication:

– The authors sort of misrepresent the field, as if the presynaptic mechanism in Kenyon cells explained all for the induction of olfactory associative memory (e.g. the beginning of the Results [L125-]). This 'presynaptic dogma' is largely based on the dispensability of Kenyon cell output during training. However, the relevant papers were published more than 20 years ago (Dubnau, Nature 2001; McGuire et al., Science 2001), and subsequent studies repeatedly reported data supporting the necessity of downstream circuits for memory induction (e.g. Krashes et al., Neuron 2007; Yamazaki et al. Cell Rep 2018). Other than these, results like the requirement of MBON output during or right after training and the importance of plasticity-related proteins therein have been published to date, all pointing to the importance of interaction with downstream circuits during training (e.g. Pai et al., PNAS 2013; Ichinose et al. Curr Biol 2021). Furthermore, live imaging strongly suggested (or claimed) for the postsynaptic mechanism (e.g. Hancock et al., Sci Rep 2022). The authors should acknowledge these studies, and modify the old presynaptic dogma.

– Imaging data on the nAChR-a2-dependent plasticity in MBONs (Figure 5) are crucial for the conclusion of this paper. The authors compared odour responses before/after training, and claimed the a2-dependent postsynaptic plasticity based on no significant differences (Figure 5h). This interpretation can be confounded, since the odour responses in these neurons drift with repeated stimulation, and this drift is different between the control and knock-down groups (Figure 5d-f). The authors need to compare the difference between paired and unpaired odour responses during the test (post-training).

– DAN activation and ACh injection at the same time caused potentiation of ACh-evoked calcium response, independent of KC (presynaptic) input (Figure 3h). On the other hand, odour response of M6 MBONs after learning showed depression (Owald et al., Neuron 2014; Figure 5g). In the light of learning-dependent depression (Figure 5), the authors should discuss and rationalize what this potentiation in Figure 3 would represent. In addition, I must point out that these 'incoherent' plasticity results are based on the calcium responses of different MBONs (M6 and M4 for Figure 3 and 5, respectively). Authors should either do the same experiment with M4 in Figure 3 (and/or M6 in Figure 5), or tone down the conclusion that the major plasticity resides in the postsynaptic sites.

– Odour-evoked calcium response was significantly increased upon knocking down the a5 subunit in M4/6 MBONs (Figure 5b and c). Why increase upon downregulating nAChR? Indeed Barnstedt et al., in Neuron 2016 showed a reduction of odour responses in the same cells. The authors should provide an explanation for this discrepancy.

– The storyline is based on different, seemingly complementary experimental batteries. However, the choices of experimental conditions are nearly random. In addition to the different cell types mentioned above, time points of measurements are very inconsistent; the behavioural experiments in Figure 1 use 30-min memory; Figure 2 uses either 3-min or 3-hr memory; imaging experiments measure immediate phenomena (Figures 3 and 5), and FRAP deals with yet other time points (Figure 6). According to their behavioural data, requirements of nAChR depend on time points after training (Figure 2). The authors should have ideally tested all under the same condition, especially if they seriously try to understand in vivo mechanisms underlying appetitive memory induction and expression (cf. title), At the very least, they should provide justification for these different conditions.

– The FRAP experiment in Figure 6 showed the reduced mobility of the a2 subunit when pairing odour and DA. A more logical explanation is needed: what this receptor mobility represents; how the reduced receptor dynamics are related to postsynaptic depression upon associative learning; why fast or slow receptor replacements should lead to different response sensitivity. In addition, receptor dynamics upon simultaneous odour+DA stimulation look just the sum of separate stimulations. If so, it may not be associative plasticity.

– In the FRAP exp (Figure 6), they bleached the GFP signal multiple times in the same sample. As the recovery is very minor (less than 10% at max), the order of the stimulations should have been randomized. Besides that, this repeated photobleaching should cause serious phototoxicity that affects their results. The authors should provide additional control experiments and/or compelling justification.

– If they promote the importance of the postsynaptic mechanism, shouldn't they also try to measure the effect of dopamine receptor knock-down in MBONs on appetitive memory? Along this line, the authors should acknowledge several papers in the field, reporting the defective appetitive and aversive memories of dopamine receptor mutants can be fully rescued in presynaptic Kenyon cells (Kim et al. J Neurosci 2007; Qin et al. Curr Biol 2009). They should also provide discussion on the role of postsynaptic dopamine input for memory induction.

– In Figure S3h, the authors showed the requirement of a2 for postsynaptic plasticity in Figure 3. As this result is important to support the postsynaptic mechanism, the authors should consider presenting it in the main figure.

– This article would gain more impact by focusing more. The main scope of this paper is about the postsynaptic plasticity for associative learning (cf. title), but they went on to show less relevant results in Figure 7 and Figure 8, explaining a mechanism of non-associative plasticity. The authors can consider reporting these results elsewhere. Alternatively, they should provide a consequent narrative to perform experiments in Figure 7 and 8.

– In my understanding, Figure 4 supports the epistatic relationship between nicotinic a5 and a2 receptors, the former being upstream of the latter via Dlg. But if so, why do they overall exhibit different phenotypes, even if they are in the same pathway (Figure 9)?

– Too much technical and procedure detail in the result part, e.g. from line 389, they should simplify the text.

– Figure 4d: it is a little weird to see a5 knockdown reduces a2 expression more than the direct knockdown of a2. The authors need to provide reasonable interpretation.

– Figure 6e: The authors need to justify the linear regression model for FRAP data.

– Figure 6d: What happens if they wait longer than 20 min? this is relevant to behavioral data as the a2 phenotype is only observable after 3 hrs (Figure 2).

– L73 "memory storage modes are functionally comparable or evolutionarily conserved."

I am not sure what the authors try to say.

---

## [Author Response]

Essential revisions:1. Please revise the manuscript focusing on and removing non-essential parts to improve readability (eg add explanations where needed for non-experts, fly genotypes,…). Also, provide a better scholarly referenced background into past work and literature. It felt sometimes that too much is trying to be said, without sufficient depth.

We have revised the text and now include several further citations, including all that reviewer 3 asked for and beyond. We would like to apologize for not including these references before – we previously clearly did not meet the standard we would like to set for ourselves, but hope to have redeemed ourselves.

As asked for, we now include explanations and definitions for wordings as well as fly genotypes in brackets and have simplified the asked for passages removing unnecessary technical details.

2. An important point that we would want you to address is the second point of Reviewer 3's list of comments where they ask to compare the difference between paired and unpaired odour responses during the test. Please also consider points 3, 5, and 7 of the same reviewer, but they may be addressed textually.

We have performed the experiments asked for: as you will see, only the paired control settings led to a depression in odor responses after training. As required, we have addressed all further points textually.

3. We ask you to include the necessary details regarding the CRISPR lines you included as "unpublished". This paper will effectively publish them.

We now include these details in the methods section.

4. All other points, including the clarifications asked for by reviewer 2 can be addressed textually.

Thank you again for these valid points. We have addressed all points.

Reviewer #1 (Recommendations for the authors):Overall the manuscript describes an important advance, both experimentally as well as conceptually. However, there are a few points that I think should be taken into account:

We thank the reviewer for their valid and constructive comments. We have addressed all points raised.

– To follow the experiments perform I feel it is important to describe in the Results section the genotypes used in the experiments performed. This can also be done in brackets to not disrupt the flow of the text. It is clear that some experiments may use complicated genotypes, however, I feel it's important that at least the logic of the experiments is apparent from the text (which Gal4, with UAS, which LexA or temp/opto line in which condition temp/light).

Thank you very much for pointing this out. We have added the requested details in the text.

– An important aspect of the manuscript relates to the differential use, localization, or expression of nAChR subunits. While the knock-down experiments described in figure 1 are easy to follow there is no information on how the expression was assessed, apart from the note that CRISPR was used as unpublished data. The authors should decide if they either want to include this set of experiments and then publish these lines or if they prefer not to publish these lines, but should then exclude this set of experiments. Without information about the locus, what has been modified, and what impact this may have it is not possible to assess these experiments properly. I am admittedly a bit dissatisfied by this section, particularly because of its central position in the logic.

We totally agree and now add the required information to the methods section (line 978ff). All of the insertion sites are chosen based on the previously successfully implemented insertion site for tagging UAS-α7GFP that has been widely-used over the last years (intracellular loop, please compare Raghu et al., 2009). We hope that the provided details now make this section and experiments better assessable. We have also added a schematic of the insertion sites to this letter (Author response image 1).

**Author response image 1. sa2fig1:** EGFP insertion sites (green arrows).

– The explant optogenetic/thermogenetic/pharmacology experiments are intriguing and – to be frank- quite refreshing. While the authors rightfully point out some limitations am sure the authors are aware of the points that could have been addressed. I do not feel that it is necessary to perform additional experiments here (though of course I'd be curious about the results), however, I would encourage the authors to at least highlight some of the obvious steps that will quite certainly allow probing their model (e.g receptors, dopamine, etc).

Thank you for pointing this out. We now have added the following to the discussion:

Line 619-626:

“Previous studies have shown that loss of DopR (dDA1) causes aversive and appetitive

memory impairments59,60. Intriguingly, specifically re-expressing DopR in KCs rescued loss of both types of memory59. However, while the reported memory impairments in dopR deficient animals were strong for aversive memories, they were only partial for appetitive memories, indicating that appetitive memory traces could be mediated via other dopamine receptors at the MBON level. Future experiments will need to investigate which dopamine receptors are required at the level of the MBONs as well as the in vivo time course of dopaminergic signaling.”

We would like to thank the reviewer for their valid and constructive criticism.

Reviewer #2 (Recommendations for the authors):As somebody not in the learning and memory field, nor with expertise in the structure and function of the insect mushroom body, I cannot provide a very helpful critique of their specific experiments. However, there are several points, from an outside point of view, which may help improve the presentation. In no particular order:

We thank the reviewer for their comments – these were very valuable and we believe have contributed a lot to improving our manuscript.

1) I think the authors should be more precise and explanatory in the terms they use. For example, what is the difference between relative familiarity with odors (non-associative familiarity learning) and habituation? Similarly, it would help if the paradigms for immediate, intermediate/3 hours, induction, and expression of memories are more clearly explained (perhaps even through schematics in one of the figures. This becomes important when they try to discuss these different forms with activities of different receptor subunits and at different MB compartments.

These are very valid and important criticisms and we thank the reviewer for pointing theses out.

1) Difference between habituation and familiarity learning: First of all, familiarity learning can indeed be seen as a specific form of habituation. To our understanding the neural processes underlying habituation can take place at several steps of the olfactory

pathway, including sensory neurons or projection neurons (that connect sensory neurons and mushroom-body intrinsic Kenyon cells). Familiarity learning, however takes place at the neural center that ‘stores’ memories. This becomes important in light of the overall mushroom body signalling logic. First, odor information is expanded from 150 projection neurons to approximately 2200 Kenyon cells per hemisphere. Each odor is represented by activity of approximately 5% of the Kenyon cells. This creates a large odor coding space, allowing for associating the odor with positive or negative reinforcers (or contexts), but also for storing information about the odor. For instance, is the odor ‘novel’ and could potentially come with a risk, or familiar? These memories, appetitive, aversive or e.g. familiarity are stored as synaptic weight changes between KCs and MBONs and prevail for certain time periods ranging from minute to days. At this step information is condensed, with the 2200 Kenyon cells converging on roughly 40 MBONs. In a simple (likely too simple!) model of downstream integration, the output of the MBON channels is summed up. Depending on the compartment the MBON gets input from, activity of the MBONs can promote approach or avoidance behaviour.

Therefore, information on novelty is likely integrated with prior experiences of encountering an odour in the context of an aversive or appetitive stimulus and all these ‘learnt’ factors are integrated allowing the animal to make a decision on which behaviour to select. So, in our view, habituation is a broader term, where plasticity can take place at several synaptic relay stations, including sensory neurons. If plasticity takes place at sensory neurons (for instance to ‘protect’ the system from sensory overload), this will affect all output channels of the MBs including those responsible for storing appetitive and aversive memories. Familiarity learning (as described by Hattori et al., 2017), however will affect a specific MBON output channel. Or as they put it: ‘The transition from novelty to familiarity is associated with suppression of neural responses in higher brain centers that appears distinct from intrinsic or sensory adaptation.’

To make this more clear, we now write: line 82ff: ‘The weights of Kenyon cells (KCs) to MB output neuron (MBON) synapses are modulated by dopaminergic neurons (DANs), which anatomically divide the MBs into at least 15 functional compartments, where information is stored on appetitive and aversive associations, in addition to non-associative information, such as the relative familiarity of an odor5,7-12,15,22,23, a distinct form of habituation. Summed up5, output from the individual MB compartments will give rise to specific behaviors, weighing up appetitive and aversive associations as well as, for instance, the familiarity of a stimulus.’

line 436ff: ‘Our data so far suggest that regulation of α2 subunits downstream of α5 are involved in postsynaptic plasticity mechanisms underlying appetitive, but not aversive memory storage. Besides associative memories, non-associative memories, such as familiarity learning, a form of habituation, are also stored at the level of *Drosophila* MBs. We next asked whether postsynaptic plasticity expressed through α5 and α2 subunit interplay, was exclusive to appetitive memory storage, or would represent a more generalizable mechanism that could underlie other forms of learning represented in the MBs.’

line 679ff: ‘We show that familiarity learning, a specific form of habituation that takes place at a higher order integration center, the mushroom bodies, can take place when knocking down α2 nAChR subunits in α’3 MBONs in principle (Figure 8), however, at clearly decreased efficacy and only after several trials. We speculate that the observation of memories still being expressed per se in this context, could be explained by redundancies with α1 or other subunits (but see heterogeneous localization and enrichment in different MB compartments, Figure 4). Redundancies could also explain why we partially observe functional phenotypes after knock-down of individual subunits, but only moderate structural changes. We also would like to point out that subunits we did not identify as absolutely required for memory expression (Figure 2) in this study could nonetheless partake in distinct phases of plasticity processes.’’

As a more general comment: we actually think that the field is indeed sometimes missing clear cut definitions – nomenclature between ‘synapse physiology’ and ‘psychology’ does not always fully match, as the ‘precision’ of localizing the mechanistic origins of observed phenomena is dimensions apart. Also, the read-outs between behaviour and physiology can be quite different. At the behavioural level, we’d agree that familiarity learning is difficult to distinguish from the broader definition of habituation. At the circuit and synapse physiological level the site of memory storage and change is decisive, which could actually warrant categorizing these separately (at least in our opinion).

2) Paradigms/schematics for induction/expression of memories: We have followed the reviewer’s advice and now include redesigned schematics for figures 1-3 and 5-8. In addition, we have added an own section in our summary figure, to visualize why each paradigm and time point was chosen. We also have added the following paragraph to

the discussion: line 664ff: ‘In order to dissect distinct roles for receptor plasticity in memory induction and expression, we experimentally probed several time points during associative appetitive memory formation. First, we probed 30 minute memory following KC block, to invariantly interfere with the memory acquisition and not the retrieval stage (Figure 1). Second, we probed immediate and 3 hour memory performance following receptor knock-down, to distinguish between memory induction and memory expression requirements (Figure 2). Third, we investigate the time course of receptor dynamics during memory expression following memory induction with a resolution of 10 minute intervals after artificial training (Figure 6, 7). The overarching picture indicates that, indeed, directly following training, memory induction requires α5. Subsequently, at the resolution of minutes, regulation of α2 levels contributes to memory expression. While we cannot resolve the temporal time course at the level of T- maze behavior (Figure 2) or FRAP experiments (Figure 6) below several minutes, our in vivo training data (Figure 5) suggests that α2 requirement already becomes apparent within 1 minute after training.’

We hope to have now presented our experimental design and rational more clearly.

2) A schematic of the MB would definitely help, with a focus on the cellular aspects and the different compartments. Their broad drawings of KC and MBONs do not help, since the work emphasizes the sub-cellular nature of receptor dynamics in specific compartments with functional specializations. Similarly, in Supplementary Figure 9, they provide EM data, but this would be helped by a cellular level that shows soma, axons/dendritic projections.

We thank the reviewer for this great idea. We have redesigned our scheme in Figure 9 and now emphasize the subcellular level (mid panels). We also show an MBON overview as suggested as panel (a) of the new Figure 10.

3) How is synaptic weight defined? And how is their plasticity defined, since they use calcium transients as opposed to actual structural changes in the synapse? Many use GCamp to monitor calcium transient changes – do all these then imply "plasticity"?

We define synaptic weight as ‘activity-dependent modification of the strength or efficacy of synaptic transmission at preexisting synapses’ (Citri and Malenka, 2007). As Citri and Malenka note in the same paper this has been: ‘proposed to play a central role in the capacity of the brain to incorporate transient experiences into persistent memory traces’. Synaptic plasticity therefore refers to a lasting change in synaptic efficacy that happens in response to a certain event (e.g. paired odor-evoked activity with dopamine release). We have reworked the text to be more precise on this account.

We have added several definitions throughout the text to make this more clear, e.g. :

Line 58ff: ‘The efficacy of synaptic transmission, also referred to as synaptic weight, can be increased or decreased following changes in neural activity profiles or concurrent action of neuromodulators, such as dopamine. Resulting changes to how synapses relay information underly synaptic plasticity, which is widely believed to be the basis of memory storage1-3. While, it is generally accepted that synaptic plasticity can serve as memory substrate from flies to humans, it is unclear to what degree neurophysiological and molecular principles underlying synaptic plasticity are evolutionarily conserved.’

And line 68ff: ‘Furthermore, it is widely established that invertebrate nervous systems utilize presynaptic plasticity (with plasticity referring to changes leading to either a strengthening (potentiation) or a weakening (depression) of synaptic transmission and the rearrangement or exchange of synaptic molecules underlying changed transmission) for storing memories, while the degree to which postsynaptic plasticity can be used is less clear.’

According to our definition changes in the strength of synapses, i.e. the amount of information transmitted at a connection would inherently underly a form of plasticity. However, depending on the type of synapse and way information is stored, synaptic plasticity can be longer lived (minutes to days) or shorter (seconds time scale). The changes reported in our manuscript deal with synaptic weight changes immediately after training that at the molecular level appear to persist at the minutes (to potentially hours) timescale.

Would every change in synaptic calcium transients be equal to a form of plasticity? We would argue that especially if triggered by an event (in our case: applied protocols that lead to a learning process) and if the change is lasting, yes. Plasticity, in our understanding, will always imply a change in functional connectivity that can alter the outcome of neural computation. We have reworked the text and only refer to plasticity where this definition is met.

4) There are many cases where different terms are used, and it is not clear to what extent these are similar words for the same process or different processes. For example, in line 272 there is this: "…postsynaptic sensitivity can change independently of the presynapse. Indeed, the observed postsynaptic potentiation was not observed…, consistent with postsynaptic plasticity…". Thus the word postsynaptic is followed by three different terms (sensitivity, potentiation, and plasticity). This is difficult for non-experts to understand. As a further example, in line 342, it would help if they explained "pre-potentiated synaptic transmission" versus "baseline transmission".

We thank the reviewer for spotting this. We have carefully revised the text and made sure to be more accurate.

For example, we have now reworked the sentences highlighted to:

line 294ff: ‘However, our proof of principle experiments demonstrate that postsynaptic plasticity at the level of MBONs can take place independently of the presynapses of the KCs. Intriguingly, we did not observe changes in calcium transient magnitudes when knocking-down α2 (UASnAChRRNAi) in M4/6 (MB011B-Split Gal4; Figure 3h), which is consistent with postsynaptic plasticity being linked to the requirement of nicotinic receptors in memory storage.’

And

line 366ff: ‘Odor responses following α5 knock-down (VT1211-Gal4>UAS-α5RNAi), however, clearly depressed after multiple odor exposures (Figure 5f), indicating that loss of α5, in comparison to the controls, actually leads to synapses being potentiated (synaptic weights are already high) from the start, even prior to the application of odors.’

5) The Discussion part is divided into seven subheadings---this seems a bit lot in terms of what the major discussion points are. Seven major take-home messages seem like a lot to emphasize for a single paper. It may help to make this part a bit more concise.

We totally agree and have cut the major take home messages to:

1) Postsynaptic plasticity in associative memory storage

2) Postsynaptic plasticity at the KC to MBON synapse

3) Nicotinic receptors could follow defined temporal sequences to mediate memory

expression

4) Are cholinergic and glutamatergic synapses interchangeable?

To make the discussion more concise, we cut some paragraphs, however also needed to add few new paragraphs in response to comments raised by the other reviewers.

However, we do hope that editing the text has now streamlined our argumentation.

6) The supplement to Figure 9 (EM data) has nothing to do with Figure 9 itself (a speculative model). Maybe the EM data can be put as part of regular Figure 9.

We totally agree, we now include the former Supplementary Figure 9 as Figure 10 in the main text and have expanded Figure 9 in response to the reviewer’s recommendations.

All in all, I think the paper is very difficult to understand for non-experts because there are so many different terms used and it's not clear what the precise differences are, whether these are fundamentally different concepts or just different terms for similar things.

We hope that by addressing the reviewer’s very valuable comments and carefully editing the text, we have improved accessibility also for non-experts.

Also, there are numerous very precise statements and not enough references at appropriate places to back them up. For example, line 504: "It also argues against the assumption that appetitive and aversive memories will necessarily use the same molecular machinery to store information. Findings in the past, predominantly based on investigating aversive memories have been generalized to learning per se". On which/whose works are such assumptions based?

We have removed this unreferenced sentence and carefully worked through the text now making sure to have added references at the appropriate places.

All in all, we would like to thank the reviewer once again for their valuable input.

Reviewer #3 (Recommendations for the authors):While the results are certainly interesting, I have a number of points for the authors to clarify before publication:

Again, we would like to thank the reviewer for their insightful and constructive comments.

– The authors sort of misrepresent the field, as if the presynaptic mechanism in Kenyon cells explained all for the induction of olfactory associative memory (e.g. the beginning of the Results [L125-]). This 'presynaptic dogma' is largely based on the dispensability of Kenyon cell output during training. However, the relevant papers were published more than 20 years ago (Dubnau, Nature 2001; McGuire et al., Science 2001), and subsequent studies repeatedly reported data supporting the necessity of downstream circuits for memory induction (e.g. Krashes et al. Neuron 2007; Yamazaki et al. Cell Rep 2018). Other than these, results like the requirement of MBON output during or right after training and the importance of plasticity-related proteins therein have been published to date, all pointing to the importance of interaction with downstream circuits during training (e.g. Pai et al. PNAS 2013; Ichinose et al. Curr Biol 2021). Furthermore, live imaging strongly suggested (or claimed) for the postsynaptic mechanism (e.g. Hancock et al. Sci Rep 2022). The authors should acknowledge these studies, and modify the old presynaptic dogma.

We thank the reviewer for their advice and would like to apologize for over-emphasizing the presynaptic dogma, and in particular for not mentioning some crucial references. We have now toned down the ‘presynaptic dogma’ and acknowledge the suggested studies along with a couple of further references that we, after careful review, felt needed inclusion. The reviewer is totally right that we previously missed to cite the study by Yamazaki et al., demonstrating the requirement for γ KC output during training – we now cite this study and have toned down our claim.

We have now modified the text accordingly throughout. Here are some examples of the changed text addressing the reviewer’s advice:

Results section, line 136ff: ‘If the postsynapse need not see the neurotransmitter during training, it would likely be dispensable for memory induction. One key argument in favor of exclusively presynaptic memory storage mechanisms in *Drosophila* is based on experiments suggesting that blocking KC or KC subset output selectively during learning leads to unaltered or mildly changed memory performance^28-31^. However, other studies have reported memory impairments following KC subset or downstream circuit element block during training^29,39,40^. Moreover, protein synthesis was shown to be required at the level of MBONs for long-term memory formation^18,41,42^.’

Discussion, line 522ff: ‘Moreover, several studies indicated that block of KCs during learning does not interfere with memory performance^28,30,31^. However, other studies blocking KC subsets did find impairments^29,39,52^ in the context of short-term appetitive memory, while downstream circuit elements have been implicated in appropriate memory formation ^18, 41, 42^.’

And

line 539ff: ‘However, a recent study investigating postsynaptic calcium transients across MBcompartments could be in line with postsynaptic modifications occurring following aversivetraining in some MB output compartments 54. Additionally, the requirement of MBON signaling has been demonstrated, particularly in the context of longer-term memory storage^18,40-42^. Thus, we do not wish to exclude a potential involvement of postsynaptic plasticity in aversive memory formation per se. On the contrary, it is conceivable that aversive memories also could have an appetitive component (release from punishment).’

While we are very thankful to the reviewer for pointing out that we did not cover the literature to the needed extent, and do recognize the clear need for altering the ‘presynaptic dogma’, we would also like to briefly mention that, to our knowledge, our manuscript is the first to actually address modes of postsynaptic memory storage in the MB.

– Imaging data on the nAChR-a2-dependent plasticity in MBONs (Figure 5) are crucial for the conclusion of this paper. The authors compared odour responses before/after training, and claimed the a2-dependent postsynaptic plasticity based on no significant differences (Figure 5h). This interpretation can be confounded, since the odour responses in these neurons drift with repeated stimulation, and this drift is different between the control and knock-down groups (Figure 5d-f). The authors need to compare the difference between paired and unpaired odour responses during the test (post-training).

We thank the reviewer for this suggestion and have repeated all experiments, this time

including unpaired controls. They are now included as part of Figure 5 (analyses of area under the curve) and Supplementary Figure 5 (analyses of peak responses). Synaptic depression is only observed in control animals using paired training and in none of the other groups.

To further improve readability and facilitate comparisons to odor responses without any learning protocols, we have decided to now present MCH responses in the main figure and octanol responses in the supplement (which also fits to the data shown in Figure 6).

– DAN activation and ACh injection at the same time caused potentiation of ACh-evoked calcium response, independent of KC (presynaptic) input (Figure 3h). On the other hand, odour response of M6 MBONs after learning showed depression (Owald et al., Neuron 2014; Figure 5g). In the light of learning-dependent depression (Figure 5), the authors should discuss and rationalize what this potentiation in Figure 3 would represent. In addition, I must point out that these 'incoherent' plasticity results are based on the calcium responses of different MBONs (M6 and M4 for Figure 3 and 5, respectively). Authors should either do the same experiment with M4 in Figure 3 (and/or M6 in Figure 5), or tone down the conclusion that the major plasticity resides in the postsynaptic sites.

The reviewer is correct: for technical reasons (explant imaging with neurotransmitter application and optogenetic stimulation), we were only able to perform the proof-of-principle experiments presented in Figure 3 in M6. In Figure 5 and in Owald et al. 2015, the observed depression was observed in M4. Following the reviewer’s suggestion, we have now expanded our discussion on these findings that we think should be taken as proof-of-principle for the ability to induce plasticity without the presynapse per se only. We now explicitly refer to this in several passages of the discussion as well as the fact that we cannot apply the findings to M4 and, for example, now write:

line 595ff: ‘It should also be noted that M4, which shows depression (Figure 5), and M6 have common but also distinct physiological roles, for instance during aversive memory extinction^13^. Besides that, different temporal requirements for M4 and M6 memory expression have been reported^11^. It is therefore possible that physiological changes in the context of appetitive learning lead to different plasticity profiles in M4 and M6 neurons respectively, or that initial potentiation over time can be reverted to depression. As noted above, MBON drive is bidirectionally modifiable and has the propensity to both potentiate and depress^7,11,17,26^. It remains unclear, whether the applied protocols would elicit plasticity (and if so depression or facilitation) at the M4 dendrites, which is difficult to assess with our experimental design. In summary, the observed ex vivo plasticity trace (Figure 3) should solely be viewed as a proof of principle that postsynaptic (MBON) plasticity can take place without presynaptic (KC) contribution per se.’

and following the reviewer’s suggestion to tone down the conclusion on postsynaptic

contributions:

line 619ff: Previous studies have shown that loss of DopR (dDA1) causes aversive and

appetitive memory impairments^59,60^. Intriguingly, specifically re-expressing DopR in KCs rescued loss of both types of memory^59^. However, while the reported memory impairments in dopR deficient animals were strong for aversive memories, they were only partial for appetitive memories, indicating that appetitive memory traces could be mediated via other dopamine receptors at the MBON level. Future experiments will need to investigate which dopamine receptors are required at the level of the MBONs as well as the in vivo time course of dopaminergic signaling.

while we also state:

line 753ff: ‘It should be noted that this architecture does not exclude presynaptic plasticity mechanisms^70^ (for instance following aversive conditioning). Indeed, we would speculate that synaptic connections can be subdivided into distinct compartments on both the pre- and the postsynaptic side, potentially through transsynaptic molecules^67,71^, allowing for fine-tuned and target-dependent changes of parameters within either side of a synapse.’

– Odour-evoked calcium response was significantly increased upon knocking down the a5 subunit in M4/6 MBONs (Figure 5b and c). Why increase upon downregulating nAChR? Indeed Barnstedt et al., in Neuron 2016 showed a reduction of odour responses in the same cells. The authors should provide an explanation for this discrepancy.

We apologize for not having made this clear enough. In Barnstedt et al. we did not image

dendritic but axonal responses. While activity at the level of the axon is the decisive factor for what is transmitted downstream (particularly relevant when assessing the overall responses after learning), we here looked into dendritic response patterns. However, decreased axonal response can be derived from both increased or decreased excitation at the level of the dendrite due to potential synaptic interference (e.g., Stuart and Spruston, 2015).

In the Results section, we have now modified this section and state:

line 349ff: ‘We next focused on implications of α2 subunit knock-down on postsynaptic

function of M4/6 MBONs. Axonal calcium transients had previously been shown to be decreased following knock-down of α subunits^4^. However, depending on the overall topology of dendritic input sites, both increased or decreased postsynaptic drive could lead to changed dendritic integration properties or potential interference of synaptic inputs, resulting in reduced signal propagation^50^.

To directly test dendritic responses, we expressed UAS-GCaMP6f in M4/6 MBONs (VT1211-Gal4) (Figure 5a), and exposed the flies repeatedly to alternating puffs of the odors octanol (OCT) and MCH (Figure 5; Supplementary Figure 5).’

the discussion on this topic now reads:

line 586ff: ‘Of note, how difficult it can be to infer how dendrites compute and integrate all input channels is exemplified by the observation that high levels of odor-mediated dendritic activation after α5 knock-down (Figure 5, Supplementary Figure 5) appear to be translated to reduced axonal calcium transients4, effectively leading to decreased signal transduction within the MBON. MBONs do not appear to exhibit prominent spines on their dendrites^23^ (but see section below: Are cholinergic and glutamatergic synapses interchangeable?). Therefore, increased dendritic activation could lead to a change in membrane resistance and result in synaptic interference.’

We would like to point-out that the presented experiments set up our in vivo imaging training under the microscope experiments. Only after knowing that the general responsiveness following knock-down of α2 was not gravely affected, were we able to perform the axonal imaging (we perform axonal imaging here to assess the integrated outcome of dendritic input) experiments shown in Figure 5k-n.

– The storyline is based on different, seemingly complementary experimental batteries. However, the choices of experimental conditions are nearly random. In addition to the different cell types mentioned above, time points of measurements are very inconsistent; the behavioural experiments in Figure 1 use 30-min memory; Figure 2 uses either 3-min or 3-hr memory; imaging experiments measure immediate phenomena (Figures 3 and 5), and FRAP deals with yet other time points (Figure 6). According to their behavioural data, requirements of nAChR depend on time points after training (Figure 2). The authors should have ideally tested all under the same condition, especially if they seriously try to understand in vivo mechanisms underlying appetitive memory induction and expression (cf. title), At the very least, they should provide justification for these different conditions.

We apologize for not making the rationale for choosing these experimental settings clear enough and thank the reviewer for pointing us towards this. To make our experimental design clearer, we have now added a timeline showing the time points assessed and also point to the technical reasons to do so (revised Figure 9, bottom row). Furthermore, we now state in the discussion:

line 664ff: ‘In order to dissect distinct roles for receptor plasticity in memory induction and expression, we experimentally probed several time points during associative appetitive memory formation. First, we probed 30 minute memory following KC block, to invariantly interfere with the memory acquisition and not the retrieval stage (Figure 1). Second, we probed immediate and 3 hour memory performance following receptor knock-down, to distinguish between memory induction and memory expression requirements (Figure 2). Third, we investigate the time course of receptor dynamics during memory expression following memory induction with a resolution of 10 minute intervals after artificial training (Figure 6, 7). The overarching picture indicates that, indeed, directly following training, memory induction requires α5. Subsequently, at the resolution of minutes, regulation of α2 levels contributes to memory expression. While we cannot resolve the temporal time course at the level of T- maze behavior (Figure 2) or FRAP experiments (Figure 6) below several minutes, our in vivo training data (Figure 5) suggests that α2 requirement already becomes apparent within 1 minute after training.‘

For adding more clarity on experimental design, we have further reworked the text. For

example, we now state for Figure 1:

line 148ff: ‘We expressed the temperature-sensitive Dynamin mutant UAS-ShibireTS (Shi) at the level of KCs (R13F02-Gal4), trained animals at the restrictive temperature (32°C), and tested for memory performance at permissive temperature (23°C) 30 minutes later. These manipulations allowed us to interfere with the synaptic vesicle exo-endocycle specifically during conditioning, while reinstating neurotransmission afterwards: by choosing the 30 minute time point, we made sure to restore functional Dynamin and not to interfere with any process underlying memory retrieval.’

We have also redesigned our figures, to make clear which mushroom body output neuron we are studying for the individual experiments.

– The FRAP experiment in Figure 6 showed the reduced mobility of the a2 subunit when pairing odour and DA. A more logical explanation is needed: what this receptor mobility represents; how the reduced receptor dynamics are related to postsynaptic depression upon associative learning; why fast or slow receptor replacements should lead to different response sensitivity.

We appreciate this point raised by the reviewer and to be frank, we would like to know the answer to this ourselves. However, as the reviewer might agree, the underlying experiments to address these questions are technically very challenging and we feel beyond the scope of this manuscript. However, we are planning to address these points in future experiments and hope to report on the results in due course.

We hope that our reworked model in Figure 9 now provides a more logical insight in the potential underlying mechanisms. At the moment, we would speculate (as mentioned we now highlight the options in our revised model in Figure 9) that the receptor in- or excorporation will happen through lateral diffusion between synaptic and extrasynaptic receptor populations (for synaptic spacing please see Figure 10b). In first experiments, we actually have probed whether we could observe fluorescence recovery following odor exposure after bleaching the entire dendritic area of the MBONs (Author response image 2). Indeed, in these experiments, we did not observe any recovery. Based on this, we, at the moment, would favour a lateral diffusion model over the second option (also highlighted in Figure 9): vesicular exocytosis.

**Author response image 2. sa2fig2:** No recovery is observed after odor stimulation after bleaching of the full dendritic arbors.

We do take up related points in the discussion. For example, we state:

line 767ff: ‘At the level of M4/6, suppressed dynamics would correspond to synaptic depression, while at the level of α’3 MBONs increased dynamics may result in postsynaptic depression. Therefore, different learning rules could govern the incorporation or exchange or mobilization of receptors in or out of synapses. The precise molecular and biophysical parameters underlying these plasticity rules are currently unknown and will need to be addressed in the future. One option could include potential exchange of α2 subunits for a receptor complex with higher calcium permeability.’

And

line 692ff: ‘In the context of both familiarity learning and appetitive conditioning, odor exposure induces increased α2 subunit dynamics (Figure 6, 7) accompanying postsynaptic depression^7,15^ (Figure 7), while not or mildly affecting α5 subunits (for familiarity learning). Therefore, the same basic mechanisms, odor-induced α2 receptor dynamics, seem to express two opposed plastic outcomes in the context of associative and non-associative memories and contribute to different learning rules across MB compartments^22,63^. We speculate that α2 dynamics induced by odor in the M4/6 dendrites could be reminiscent of dark currents in the vertebrate visual system^64^ allowing for rapid adaptation with low levels of synaptic noise. Receptor exchange at the level of M4/6 dendrites would actually take place when no associations are formed and stalled when dopaminergic neurons (triggered by sugar) are simultaneously active with KCs (triggered by odor). Indeed, repeated Oct stimulation (Supplementary Figure 5) led to a facilitation of calcium transients (potentially corresponding to an increase of receptor incorporation), while depression (in this case likely to be mediated by removal of receptors, but see above) is triggered by paired training (Figure 5). In contrast, at the level of the α’3 compartments, odor activates both MBONs and dopaminergic neurons. Here, the plasticity rule would be reversed. Synaptic depression is accompanied by actively changing the receptor composite. We speculate that such plasticity could function reminiscent of mechanisms observed for climbing fiber-induced depression of parallel fiber to Purkinje cell synapses^65^. However, whether increased dynamics can be translated to more incorporation or removal of α2-type receptors, or depending on the plasticity rule both, will require high resolution imaging experiments in the future.’

In addition, receptor dynamics upon simultaneous odour+DA stimulation look just the sum of separate stimulations. If so, it may not be associative plasticity.

Our data show that odor actively triggers receptor dynamics (Figure 6 and Supplementary Figure 6).

This active trigger is suppressed by coincident DA application. As the active process of recovery induced by the odor is suppressed by DA, it is indeed the coincidence of both stimuli that manifest the change in receptor dynamics. We follow the reviewer’s argumentation that DA application (as also applying no stimulus at all (‘air only’), please see Supplementary Figure 6e) by itself does not trigger receptor recovery. However, even if the presence of DA would be the factor that always suppresses recovery (regardless whether in the presence of odor or not) it requires the coincidence of DA with the odor to change the cellular response (here receptor dynamics) triggered by an odor.

– In the FRAP exp (Figure 6), they bleached the GFP signal multiple times in the same sample. As the recovery is very minor (less than 10% at max), the order of the stimulations should have been randomized. Besides that, this repeated photobleaching should cause serious phototoxicity that affects their results. The authors should provide additional control experiments and/or compelling justification.

In order to control for position effects, we performed similar experiments as in Figure 6 (all conditions probed subsequently in the same fly) in individual flies, were we presented only one of the conditions (without any previous history, Supplementary Figure 6). As seen for the results presented in Figure 6 and Supplementary Figure 6, only odor triggered signal recovery and none of the other conditions. We can exclude position effects, as the experiments presented in Supplementary Figure 6 actually have no position. These observations from individual flies also argue against our results being a consequence of phototoxicity. We further ruled out phototoxic effects in additional experiments. First, when we started this experimental series, we measured MBON odor responses with co-expressed calcium indicators following the bleaching protocols. Arguing for healthy tissue, odor responses were clearly observable after conducting bleaching protocols (not shown; of note, these are difficult experiments as red calcium indicators bleach fast under the settings we use). Second, we performed preliminary experiments, where we exposed the flies to a novel odor (OCT) after pairing MCH and DA. As you can see from our preliminary data in Author response image 3, we observe robust recovery following the third round of bleaching, again arguing against phototoxic effects.

**Author response image 3. sa2fig3:** preliminary experiments demonstrating recovery of signal after three rounds of photo bleaching to a novel odor arguing against significant photo-toxic effects.

– If they promote the importance of the postsynaptic mechanism, shouldn't they also try to measure the effect of dopamine receptor knock-down in MBONs on appetitive memory? Along this line, the authors should acknowledge several papers in the field, reporting the defective appetitive and aversive memories of dopamine receptor mutants can be fully rescued in presynaptic Kenyon cells (Kim et al. J Neurosci 2007; Qin et al. Curr Biol 2009). They should also provide discussion on the role of postsynaptic dopamine input for memory induction.

We agree that identifying the relevant dopamine receptors at the MBON level would be highly interesting, and definitely something we would like to address in the future. For now, to us, this is beyond the scope of this paper. We have however followed the reviewer’s very justified suggestion and included a paragraph discussing previous findings.

We would like to highlight that the listed studies showed that aversive memories were fully abolished in DopR/DopR1/dDA1 mutants and rescued by KC-specific re-expression (compare Kim et al., 2007). For appetitive memories, however, the impairment of memory performance observed was partial. While this partial effect was indeed rescuable by KC expression, this leaves room for a significant component being MBON-dependent. As we also state in the discussion (line 755ff), we actually do suspect pre-and postsynaptic modifications to both take place during memory formation:

‘Indeed, we would speculate that synaptic connections can be subdivided into distinct compartments on both the pre- and the postsynaptic side, potentially through transsynaptic molecules^67,71^, allowing for fine-tuned and target-dependent changes of parameters within either side of a synapse.’

As suggested, we also now include the following paragraph to the discussion:

line 619: ‘Previous studies have shown that loss of DopR (dDA1) causes aversive and

appetitive memory impairments^59,60^. Intriguingly, specifically re-expressing DopR in KCs rescued loss of both types of memory^59^. However, while the reported memory impairments in dopR deficient animals were strong for aversive memories, they were only partial for appetitive memories, indicating that appetitive memory traces could be mediated via other dopamine receptors at the MBON level. Future experiments will need to investigate which dopamine receptors are required at the level of the MBONs as well as the in vivo time course of dopaminergic signaling.’

– In Figure S3h, the authors showed the requirement of a2 for postsynaptic plasticity in Figure 3. As this result is important to support the postsynaptic mechanism, the authors should consider presenting it in the main figure.

We thank the reviewer for pointing this out and have moved the experiment to the main figure.

– This article would gain more impact by focusing more. The main scope of this paper is about the postsynaptic plasticity for associative learning (cf. title), but they went on to show less relevant results in Figure 7 and Figure 8, explaining a mechanism of non-associative plasticity. The authors can consider reporting these results elsewhere. Alternatively, they should provide a consequent narrative to perform experiments in Figure 7 and 8.

We would prefer to keep these experiments as we feel that they convey an important biological message (several memory forms require postsynaptic receptor plasticity) and because some experiments were feasible to conduct at this more superficial area of the brain in vivo (the tip of the MB vertical lobes). If, however, keeping these data would interfere with publishing our manuscript, we are prepared to remove these data. Please also see our comment above. To justify our experiments more clearly, we include familiarity learning in title and abstract. We also have reworked the paragraph in the Results section highlighting our reasoning to perform the experiments on familiarity learning:

line 436ff: ‘Our data so far suggest that regulation of α2 subunits downstream of α5 are involved in postsynaptic plasticity mechanisms underlying appetitive, but not aversive memory storage. Besides associative memories, non-associative memories, such as familiarity learning, a form of habituation, are also stored at the level of the *Drosophila* MBs. We next asked whether postsynaptic plasticity expressed through α5 and α2 subunit interplay, was exclusive to appetitive memory storage, or would represent a more generalizable mechanism that could underlie other forms of learning represented in the MBs. We turned to the α’3 compartment at the tip of the vertical MB lobe that has previously been shown to mediate odor familiarity learning. This form of learning allows the animal to adapt its behavioral responses to new odors and permits for assaying direct odor-related plasticity at the level of a higher order integration center. Importantly, this compartment follows different plasticity rules, because the odor serves as both the conditioned (activating KCs) and unconditioned stimulus (activating corresponding dopaminergic neurons)15. While allowing us to test whether the so far uncovered principles could also be relevant in a different context, it also provides a less complex test bed to further investigate whether α5 functions upstream of α2 dynamics.’

– In my understanding, Figure 4 supports the epistatic relationship between nicotinic a5 and a2 receptors, the former being upstream of the latter via Dlg. But if so, why do they overall exhibit different phenotypes, even if they are in the same pathway (Figure 9)?

We totally agree with the reviewer. Figure 4 suggest an epistatic relationship with α5 functioning upstream of Dlg and α2. Indeed, Figure 2 also suggests that α5 is needed for memory induction and α2 for the expression phase downstream. Moreover, at the level of α’3 MBONs, cell-specific knock-down of α5 prevents α2 recovery, adding another line of evidence for an epistatic relationship (Figure 7). The reviewer is fully correct in pointing out that the phenotypes observed are not fully overlapping between α5, Dlg and α2. In our model, α5 takes the role of a gatekeeper allowing to trigger or induce downstream processes. The easiest explanation would be that upstream α5 can have several other downstream effectors besides Dlg and α2 and therefore exhibits a more pronounced and severe phenotype. However, experiments in Figure 5 also point towards loss of α5 leaving the synapse (and as a consequence behavior) in a state with high calcium responses to an odor from the first stimulation onwards. Thus, in line with a disturbed gating mechanism when downregulating α5, synapses appear to be ‘prepotentiated’, or, already in a state, at least approaching the upper barrier of transmission efficacy, that also depress rapidly, quickly approaching the lower barrier. Thus, the dynamic range of weight changes is no longer available to the synapse without α5. This is further reflected in Figure 8, where animals with α’3 MBON-specific knock-down of α5 nAChR subunits behave as if they had learned about the familiarity of an odor before (without actually having learnt anything about the odor), which is likely reflected in rapid synaptic rundown (compare Figure 5). The precise mechanisms underlying the proposed gating mechanisms as well as further downstream targets of α5 will need to be worked out in the future. However, we would like to point out that α2 shows phenotypes that are overlapping with α5 (e.g. Figure 2), which is consistent with α2 functioning downstream of α5, but α5 having more (to be identified – for instance α1, please compare Figure 2) targets.

– Too much technical and procedure detail in the result part, e.g. from line 389, they should simplify the text.

We agree and thank the reviewer for their advice. We have reduced the technical aspects in the Results section wherever it appeared feasible to us, and especially for the section on in vivo FRAP experiments highlighted by the reviewer. We must point out that we also did add some technical detail (genetics) to the main text body as requested by reviewer 1.

– Figure 4d: it is a little weird to see a5 knockdown reduces a2 expression more than the direct knockdown of a2. The authors need to provide reasonable interpretation.

We thank the reviewer for spotting this. In response to the reviewer’s observation, we

compared the α2 and α5 knockdown groups and found that they were not statistically different. Indeed, when comparing their median, both groups are comparable (please see Author response image 4).

However, what we do see is an increased range (Author response image 4) that likely originates from higher experimental noise, explaining the observed effect.

**Author response image 4. sa2fig4:** Median and range for data shown in Figure 5d.

– Figure 6e: The authors need to justify the linear regression model for FRAP data.

We agree that we had not sufficiently justified our model and have added further explanations to the methods section and figure legends. In brief, for fitting the FRAP data, we assumed an inverse exponential decay of the form 1 ⁻ e^-k*t^, where k denotes the decay constant and t time, as commonly used for FRAP data. To that end, before fitting of the linear mixed effect models, we inverted and log-transformed the intensity values, which is equivalent to fitting an inverse exponential decay function. For plotting, the predicted values were back-transformed to the original scale.

To further clarify this, we added more details to the methods section and to the figure legends:

1) Methods (line 906ff): ‘Photobleaching was accomplished using focused, high intensity laser exposure for ~1 minute. Analysis of fluorescence recovery was performed using FIJI. ROIs were manually selected and the percent recovery fluorescence was calculated by subtraction of the post-bleaching baseline fluorescence and division by the pre-bleaching baseline fluorescence. To fit the inverse exponential decay that is expected for FRAP data, we first inverted the percent fluorescence recovery values by subtracting them from 1 and then log-transformed the resulting values. These log-transformed values were used in a linear mixed effects model without intercept using the interaction between condition and time as fixed effect – to determine condition-specific differences of the recovery kinetics - and time as random effect (R package lme4). A linear mixed effect model was used to appropriately model repeated measures within animals. By inverting and log-transforming the fluorescence recovery values, this approach is equivalent to fitting an inverted exponential decay function. For plotting, all values including regression coefficients were back-transformed to the original scale. Significance of recovery of individual conditions was assessed using the regression coefficients of the condition-time interaction of the linear mixed model. Differences of recovery between pairs of conditions were tested using pairwise comparisons of estimated marginal means of the linear mixed model (R package emmeans). Correction for multiple pairwise comparisons was performed using Tukey’s method.’

2) Legend to Figure 6: ‘d)Inverse exponential decay fit of fluorescence recovery after photobleaching following MCH exposure (blue line), MCH exposure simultaneously with dopamine (DA) injection (purple line), dopamine injection alone (red line); (e) Regression coefficient for the inverse exponential decay fit. Bar graphs: regression coefficients of α2 RNAi α5 RNAi -40 -20 0 20 relative fluorescence 21 recovery kinetics ± standard error of regression; n = 9 – 10, linear mixed effects model followed by pairwise comparison from estimated marginal trends. * = p < 0.05’.

3) According edits to the legend of Supplementary Figure 6.

– Figure 6d: What happens if they wait longer than 20 min? this is relevant to behavioral data as the a2 phenotype is only observable after 3 hrs (Figure 2).

We also conducted FRAP experiments probing the 30 min time point. As can be seen from Author response image 5, the largest recovery part takes place within the first 10 minutes and subsequently appears to approach a plateau. We expect most of the plasticity mechanisms therefore to take place within the first 10 to 20 min. However, we probe memory at the later 3 hr time point, to encapsulate the whole potential ‘memory-expression’ time line. Our model therefore suggests that α2 dynamics are induced downstream of α5 after training. The changed properties are manifest in later memory phases that can be probed at the 3 hr time point. Please also see our new Figure 9 and, as also pointed out above, our new paragraph to the discussion:

line 664ff: ‘In order to dissect distinct roles for receptor plasticity in memory induction and expression, we experimentally probed several time points during associative appetitive memory formation. First, we probed 30 minute memory following KC block, to invariantly interfere with the memory acquisition and not the retrieval stage (Figure 1). Second, we probed immediate and 3 hour memory performance following receptor knock-down, to distinguish between memory induction and memory expression requirements (Figure 2). Third, we investigate the time course of receptor dynamics during memory expression following memory induction with a resolution of 10 minute intervals after artificial training (Figure 6, 7). The overarching picture indicates that, indeed, directly following training, memory induction requires α5. Subsequently, at the resolution of minutes, regulation of α2 levels contributes to memory expression. While we cannot resolve the temporal time course at the level of T- maze behavior (Figure 2) or FRAP experiments (Figure 6) below several minutes, our in vivo training data (Figure 5) suggests that α2 requirement already becomes apparent within 1 minute after training.”

**Author response image 5. sa2fig5:** α2 recovery at 10, 20 and 30 min.

– L73 "memory storage modes are functionally comparable or evolutionarily conserved."I am not sure what the authors try to say.

We apologize for not being clear and have rephrased this sentence that now reads:

line 77f: ‘Detailed knowledge of the anatomical wiring and functional signaling logic of the *Drosophila* learning and memory centers, the mushroom bodies5,^7-21^ (MBs; third (‘higher’) order brain center; learning takes place three synapses downstream of sensory neurons), allows one to address to what extent synaptic mechanisms underlying memory storage are comparable across evolution, despite the use of different neurotransmitter systems.’